# Inverse regulation of *Vibrio cholerae* biofilm dispersal by polyamine signals

**Andrew A Bridges[1,2], Bonnie L Bassler[1,2]\***

[1]Department of Molecular Biology, Princeton University, Princeton, United States; [2]The Howard Hughes Medical Institute, Chevy Chase, United States

**Abstract** The global pathogen *Vibrio cholerae* undergoes cycles of biofilm formation and dispersal in the environment and the human host. Little is understood about biofilm dispersal. Here, we show that MbaA, a periplasmic polyamine sensor, and PotD1, a polyamine importer, regulate *V. cholerae* biofilm dispersal. Spermidine, a commonly produced polyamine, drives *V. cholerae* dispersal, whereas norspermidine, an uncommon polyamine produced by vibrios, inhibits dispersal. Spermidine and norspermidine differ by one methylene group. Both polyamines control dispersal via MbaA detection in the periplasm and subsequent signal relay. Our results suggest that dispersal fails in the absence of PotD1 because endogenously produced norspermidine is not reimported, periplasmic norspermidine accumulates, and it stimulates MbaA signaling. These results suggest that *V. cholerae* uses MbaA to monitor environmental polyamines, blends of which potentially provide information about numbers of 'self' and 'other'. This information is used to dictate whether or not to disperse from biofilms.

**\*For correspondence:**
bbassler@princeton.edu

**Competing interests:** The authors declare that no competing interests exist.

## Introduction

Bacteria frequently colonize environmental habitats and infection sites by forming surface-attached multicellular communities called biofilms. Participating in the biofilm lifestyle allows bacteria to collectively acquire nutrients and resist threats (*Flemming et al., 2016*). By contrast, the individual free-swimming state allows bacteria to roam. The global pathogen *Vibrio cholerae* undergoes repeated rounds of clonal biofilm formation and disassembly, and both biofilm formation and biofilm exit are central to disease transmission as *V. cholerae* alternates between the marine niche and the human host (*Conner et al., 2016*; *Gallego-Hernandez et al., 2020*; *Tamayo et al., 2010*). The *V. cholerae* biofilm lifecycle occurs in three stages. First, a founder cell attaches to a substate; second, through cycles of growth, division, and extracellular matrix secretion, the biofilm matures; and finally, when the appropriate environmental conditions are detected, cells leave the biofilm through a process termed dispersal (*Bridges et al., 2020*). Bacterial cells liberated through dispersal can depart their current environment and repeat the lifecycle in a new niche, such as in a new host (*Guilhen et al., 2017*). In general, the initial stages of the biofilm lifecycle, from attachment through maturation, have been well studied in *V. cholerae* and other bacterial species, whereas the dispersal phase remains underexplored. Components underlying biofilm dispersal are beginning to be revealed. For example, in many bacterial species, the Lap system that is responsible for cell attachments during biofilm formation is also involved in dispersal (*Boyd et al., 2012*; *Christensen et al., 2020*; *Gjermansen et al., 2005*; *Kitts et al., 2019*; *Newell et al., 2011*).

To uncover mechanisms controlling *V. cholerae* biofilm dispersal, we recently developed a bright-field microscopy assay that allows us to follow the entire biofilm lifecycle (*Bridges and Bassler, 2019*). We combined this assay with mutagenesis and high-content imaging to identify mutants that failed to properly disperse (*Bridges et al., 2020*). Our screen revealed genes encoding proteins that fall into three functional groups: signal transduction, matrix degradation, and cell motility. Our hypothesis is that these classes of components act in sequence to drive biofilm dispersal. First, the

cues that induce dispersal are detected by the signal transduction proteins. Second, activation of matrix digestion components occurs. Finally, cell motility engages and permits cells to swim away from the disassembling biofilm. The majority of the genes identified in the screen encoded signal transduction proteins. Elsewhere, we characterized one of the signaling systems identified in the screen, a new two-component phospho-relay that we named DbfS-DbfR (for Dispersal of Biofilm Sensor and Dispersal of Biofilm Regulator) that controls biofilm dispersal (*Bridges et al., 2020*). Of the remaining signaling proteins identified in the screen, four are proteins involved in regulating production/degradation of, or responding to, the second messenger molecule cyclic diguanylate (c-di-GMP).

c-di-GMP regulates biofilm formation in many bacteria including *V. cholerae* in which low c-di-GMP levels correlate with motility and high c-di-GMP levels promote surface attachment and matrix production, and thus, biofilm formation (*Conner et al., 2017*). c-di-GMP is synthesized by enzymes that contain catalytic GGDEF domains that have diguanylate cyclase activity (*Conner et al., 2017*). c-di-GMP is degraded by proteins with EAL or HD-GYP domains that possess phosphodiesterase activity. *V. cholerae* encodes >50 proteins harboring one or both of these domains, underscoring the global nature of c-di-GMP signaling in this pathogen. The activities of these enzymes are often regulated by environmental stimuli through ligand binding sensory domains; however, in most cases, the identities of ligands for GGDEF- and EAL-containing enzymes are unknown (*Römling et al., 2013*). Of the c-di-GMP regulatory proteins identified in our dispersal screen, only one, the hybrid GGDEF/EAL protein, MbaA, which has previously been shown to regulate *V. cholerae* biofilm formation (*Bomchil et al., 2003*), has a defined environmental stimulus: MbaA responds to the ubiquitous family of small molecules called polyamines, aliphatic cations containing two or more amine groups (*Miller-Fleming et al., 2015*). Our dispersal screen also identified PotD1, a periplasmic binding protein that mediates polyamine import and that has also been shown to control *V. cholerae* biofilm formation (*Cockerell et al., 2014*; *McGinnis et al., 2009*). Thus, identification of these two genes in our screen suggested that beyond roles in biofilm formation, polyamine detection could be central to biofilm dispersal. In the present work, we explore this new possibility.

Polyamines are essential small molecules derived from amino acids and were likely produced by the last universal common ancestor (*Michael, 2018*; *Miller-Fleming et al., 2015*). Different organisms make distinct blends of polyamines (*Michael, 2018*), and they are involved in biological processes ranging from stress responses to signal transduction to protein synthesis to siderophore production (*Karatan et al., 2005*; *McGinnis et al., 2009*; *Michael, 2018*). In some cases, mechanisms underlying polyamine function are known. For example, the eukaryotic translation initiation factor 5A is modified with hypusine that is derived from the polyamine spermidine (*Puleston et al., 2019*). In most cases, however, the mechanisms by which polyamines control physiology remain mysterious (*Miller-Fleming et al., 2015*).

Polyamines are known to be involved in biofilm development in many bacterial species (*Hobley et al., 2014*; *Karatan and Michael, 2013*; *Michael, 2018*; *Nesse et al., 2015*; *Patel et al., 2006*). Intriguingly, the biofilm alterations that occur in response to a given polyamine vary among species. In *V. cholerae*, norspermidine, a rare polyamine produced by Vibrionaceae and select other organisms, promotes biofilm formation (*Karatan et al., 2005*; *Lee et al., 2009*). In contrast, the nearly ubiquitous polyamine spermidine, which differs from norspermidine only by a single methylene group, is not produced at substantial levels by *V. cholerae* and it represses *V. cholerae* biofilm formation (*Figure 1*; *Karatan et al., 2005*; *Lee et al., 2009*; *McGinnis et al., 2009*). Likewise, the polyamine spermine, typically made by eukaryotes, also represses *V. cholerae* biofilm formation (*Sobe et al., 2017*). Both spermidine and spermine are abundant in the human intestine (*Benamouzig et al., 1997*; *McEvoy and Hartley, 1975*; *Osborne and Seidel, 1990*). These results have led to the hypothesis that *V. cholerae* detects norspermidine as a measure of 'self' and spermidine and spermine as measures of 'other' to assess the species composition of the vicinal community (*Sobe et al., 2017*; *Wotanis et al., 2017*). *V. cholerae* funnels that information internally to determine whether or not to make a biofilm (*Sobe et al., 2017*; *Wotanis et al., 2017*). *V. cholerae* produces intracellular norspermidine; however, norspermidine has not been detected in cell-free culture fluids of laboratory-grown strains (*Parker et al., 2012*). Thus, if norspermidine does indeed enable *V. cholerae* to take a census of 'self,' it is apparently not via a canonical quorum-sensing-type mechanism.

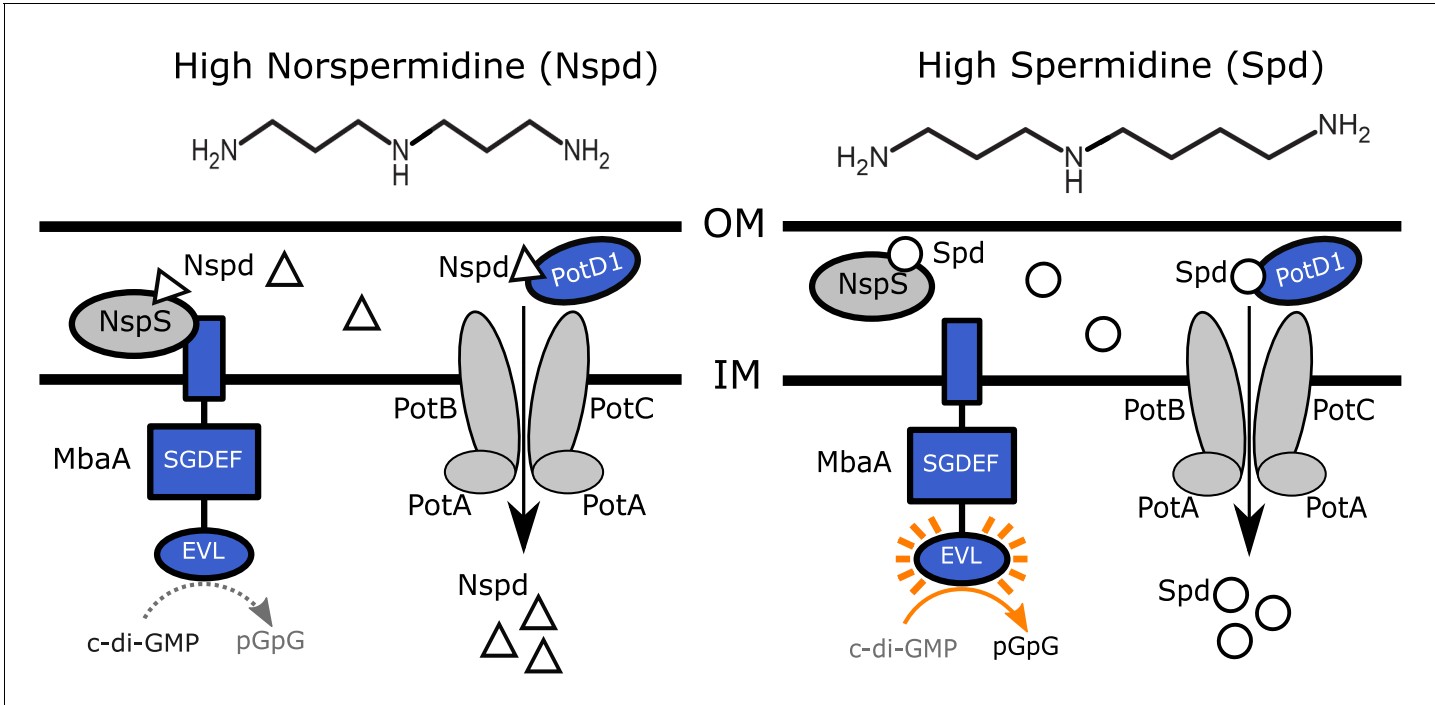

**Figure 1.** Polyamine sensing in *V. cholerae*. Schematic showing the previously proposed polyamine detection and import mechanisms in *V. cholerae*. Norspermidine (Nspd, triangles) promotes biofilm formation and spermidine (Spd, circles) represses biofilm formation. The primarily eukaryotic polyamine spermine (not pictured) also signals through the NspS-MbaA pathway. See text for details. OM: outer membrane; IM: inner membrane.

As mentioned above, information contained in extracellular polyamines is transduced internally by MbaA, which is embedded in the inner membrane. MbaA contains a periplasmic domain that interacts with the periplasmic binding protein NspS. *nspS* is located immediately upstream of *mbaA* in the chromosome (*Cockerell et al., 2014*; *Karatan et al., 2005*; *Young et al., 2021*). MbaA also possesses cytoplasmic GGDEF (SGDEF in MbaA) and EAL (EVL in MbaA) domains (*Figure 1*). Genetic evidence suggests that when NspS is bound to norspermidine, the complex associates with MbaA and biofilm formation is promoted (*Cockerell et al., 2014*). Specifically, NspS interaction with MbaA inhibits MbaA phosphodiesterase activity, leading to increased c-di-GMP levels, and in turn, elevated biofilm formation (*Figure 1*, left panel) (*Cockerell et al., 2014*). It is proposed that Apo-NspS and NspS bound to spermidine or spermine do not bind to MbaA, and thus, under this condition, biofilm formation is reduced (*Cockerell et al., 2014*; *Sobe et al., 2017*). In this case, MbaA phosphodiesterase activity is not inhibited, which leads to decreased c-di-GMP levels and repression of biofilm formation (*Figure 1*, right panel). Currently, it is not known whether or not MbaA has diguanylate cyclase activity. The purified MbaA cytoplasmic domain functions as a phosphodiesterase in vitro (*Cockerell et al., 2014*).

In a proposed second regulatory mechanism, norspermidine and spermidine are thought to control biofilm formation via import through the inner membrane ABC transporter, PotABCD1 (*McGinnis et al., 2009*). Following internalization, via an undefined cytoplasmic mechanism, polyamines are proposed to modulate biofilm formation (*Figure 1*; *McGinnis et al., 2009*). This hypothesis was based on the finding that elimination of PotD1, which is required for import of norspermidine and spermidine, resulted in elevated biofilm formation. Importantly, the studies reporting the MbaA and PotD1 findings used quorum-sensing-deficient *V. cholerae* strains (*Joelsson et al., 2006*) that likely cannot disperse from biofilms. Furthermore, the end-point biofilm assays used could not differentiate between enhanced biofilm formation and the failure to disperse.

Here, we combine real-time biofilm lifecycle measurements, mutagenesis, a reporter of cytoplasmic c-di-GMP levels, and measurements of intra- and extracellular polyamine concentrations to define the roles that norspermidine and spermidine play in controlling the *V. cholerae* biofilm lifecycle. In wildtype *V. cholerae*, exogenous norspermidine and spermidine inversely alter cytoplasmic

c-di-GMP levels and they exert their effects through the NspS-MbaA circuit. Norspermidine promotes biofilm formation and suppresses biofilm dispersal. Spermidine represses biofilm formation and promotes biofilm dispersal. Both the MbaA SGDEF and EVL domains are required for polyamine control of the biofilm lifecycle. We provide evidence that MbaA synthesizes c-di-GMP in the presence of norspermidine and degrades c-di-GMP in the presence of spermidine and that is what drives *V. cholerae* to form and disperse from biofilms, respectively. When MbaA is absent, *V. cholerae* is unable to alter the biofilm lifecycle in response to extracellular polyamines. We demonstrate that polyamine internalization via PotD1 is not required to promote *V. cholerae* entrance or exit from biofilms, but rather, periplasmic detection of polyamines by MbaA is the key regulatory step. Specifically, our results suggest that the Δ*potD1* mutant fails to disperse because it is unable to reimport self-secreted norspermidine. The consequence is that excess periplasmic norspermidine accumulates, is detected by MbaA, and leads to production of c-di-GMP and suppression of biofilm dispersal. Collectively, our work reveals the mechanisms by which *V. cholerae* detects and transduces the information contained in polyamine signals into modulation of its biofilm lifecycle. We propose that the polyamine sensing mechanisms revealed in this study allow *V. cholerae* to distinguish relatives from competitors and potentially the presence of predators in the vicinity and, in response, modify its biofilm lifecycle to appropriately colonize territory or disperse from an existing community.

## Results

### The polyamine signaling proteins MbaA and PotD1 regulate biofilm dispersal by changing c-di-GMP levels

Our combined mutagenesis-imaging screen identified the inner membrane polyamine sensor, MbaA, and the periplasmic polyamine binding protein PotD1 as essential for proper *V. cholerae* biofilm dispersal, motivating us to explore the mechanisms underlying these effects. To probe polyamine signaling across the full biofilm lifecycle, we used our established brightfield imaging assay. In the case of WT *V. cholerae*, peak biofilm biomass is reached at ~8–9 hr of growth, and subsequently, dispersal occurs, and is completed by ~12 hr post inoculation (*Figure 2A, B*). Deletion of *mbaA* caused a mild biofilm dispersal defect with no detectable difference in peak biofilm biomass compared to WT, a less than 1 hr delay in the onset of dispersal, and 27% biomass remaining at 16 hr (*Figure 2A, B*). Expression of *mbaA* from an ectopic locus in the Δ*mbaA* mutant complemented the biofilm dispersal defect (*Figure 2—figure supplement 1A*). Thus, MbaA is required for WT *V. cholerae* biofilm dispersal. The Δ*potD1* mutant exhibited a 60% greater peak biofilm biomass than WT and nearly all of the biomass remained at 16 hr, indicating that PotD1 both represses biofilm formation and promotes biofilm dispersal (*Figure 2A, B*). Introduction of *potD1* at an ectopic locus in the Δ*potD1* mutant drove premature biofilm dispersal and reduced overall biofilm formation (*Figure 2—figure supplement 1B*). The differences in severity between the dispersal phenotypes of the Δ*mbaA* and Δ*potD1* mutants are noteworthy because these strains behaved similarly in the previous endpoint biofilm formation assays (*McGinnis et al., 2009*).

MbaA has previously been shown to possess phosphodiesterase activity in vitro (*Cockerell et al., 2014*), thus, we reasoned that MbaA functions by altering cytoplasmic c-di-GMP levels, the consequence of which is changes in expression of genes encoding biofilm components. As far as we are aware, no connection has yet been made between PotD1 and cytoplasmic c-di-GMP levels. To determine if the observed biofilm dispersal defects in the Δ*mbaA* and Δ*potD1* mutants track with altered c-di-GMP levels, we compared the relative cytoplasmic c-di-GMP levels in the WT, Δ*mbaA*, and Δ*potD1* strains. To do this, we employed a fluorescent reporter construct in which expression of *turboRFP* is controlled by two c-di-GMP-responsive riboswitches (*Zhou et al., 2016*). The TurboRFP signal is normalized to constitutively produced AmCyan encoded on the same plasmid. Previous studies have demonstrated a linear relationship between reporter output and c-di-GMP levels measured by mass spectrometry (*Zhou et al., 2016*). Indeed, the reporter showed that deletion of *mbaA* caused only a moderate increase in cytoplasmic c-di-GMP (8% higher than WT), while the Δ*potD1* mutant produced 39% more signal than WT (*Figure 2C*). Thus, MbaA, as expected, mediates changes in c-di-GMP levels, and moreover PotD1-mediated import of polyamines also influences cytoplasmic c-di-GMP concentrations.

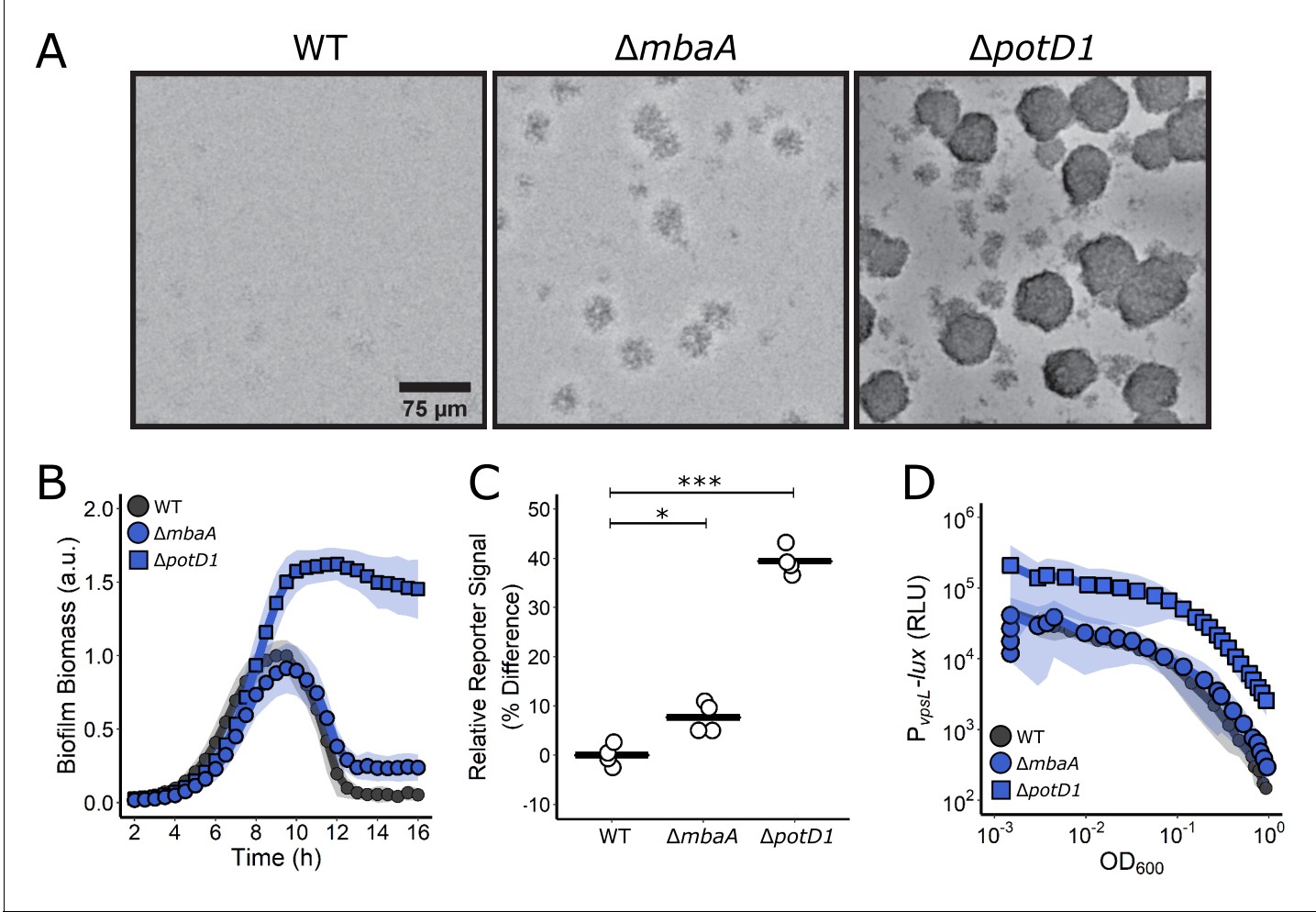

**Figure 2.** Polyamine signaling regulates *V. cholerae* biofilm dispersal. (**A**) Representative images of the designated *V. cholerae* strains at 16 hr. (**B**) Quantitation of biofilm biomass over time measured by time-lapse microscopy for WT *V. cholerae* and the designated mutants. In all cases, $N = 3$ biological and $N = 3$ technical replicates, ± SD (shaded). a.u.: arbitrary unit. (**C**) Relative c-di-GMP reporter signals for the indicated strains. Values are expressed as the percentage difference relative to the WT strain. $N = 4$ biological replicates. Each black bar shows the sample mean. Unpaired t-tests were performed for statistical analysis, with p values denoted as *$p<0.05$; ***$p<0.001$. (**D**) The corresponding *PvpsL-lux* outputs for the strains and growth conditions in (**B**). For *vpsL-lux* measurements, $N = 3$ biological replicates, ± SD (shaded). RLU: relative light units.

The online version of this article includes the following figure supplement(s) for figure 2:

**Figure supplement 1.** *mbaA* and *potD1* complement the Δ*mbaA* and Δ*potD1* mutants, respectively.

**Figure supplement 2.** c-di-GMP signaling controls biofilm dispersal.

In *V. cholerae*, increased cytoplasmic c-di-GMP levels are associated with elevated extracellular matrix production (called VPS for vibrio polysaccharide) and, in turn, increased biofilm formation. Using a $P_{vpsL}$-*lux* promoter fusion that reports on the major matrix biosynthetic operon, we previously showed that matrix gene expression decreases as cells transition from the biofilm to the planktonic state, suggesting that repression of matrix production genes correlates with biofilm dispersal. We wondered how the increased c-di-GMP levels present in the Δ*mbaA* and Δ*potD1* mutants impinged on *vpsL* expression. Using the *vpsL-lux* reporter, we found that the light production patterns mirrored the severities of the dispersal phenotypes and the magnitudes of changes in cytoplasmic c-di-GMP levels: the Δ*mbaA* mutant had a light production profile similar to WT, while the Δ*potD1* mutant produced 10-fold more light than WT throughout growth (*Figure 2D*). These results indicate that the Δ*mbaA* mutant makes normal levels and the Δ*potD1* mutant produces excess matrix.

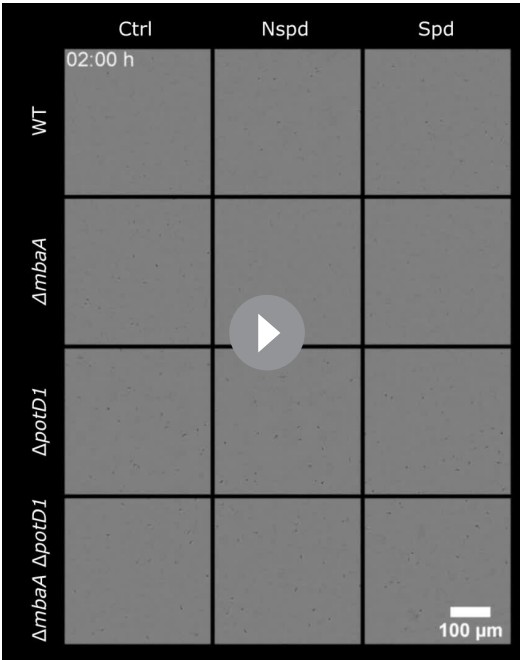

**Video 1.** Representative time-lapse images of the biofilm lifecycles of the WT, Δ*mbaA*, Δ*potD1*, and Δ*mbaA* Δ*potD1* *V. cholerae* strains following treatment with water (Ctrl), 100 μM norspermidine (Nspd), or 100 μM spermidine (Spd).

https://elifesciences.org/articles/65487#video1

As mentioned in the Introduction, our screen revealed three additional genes in c-di-GMP signaling pathways involved in *V. cholerae* biofilm dispersal. While not the focus of the current work, we performed preliminary characterization by deleting these genes and making measurements of the mutants' biofilm lifecycle phenotypes, assessing them for changes in cytoplasmic c-di-GMP levels, and quantifying their matrix production profiles (*Figure 2—figure supplement 2A–D*).

## Norspermidine and spermidine inversely regulate *V. cholerae* biofilm dispersal

Above, we show that polyamine signaling proteins are required for normal *V. cholerae* biofilm dispersal. Previous studies demonstrated that norspermidine and spermidine have opposing effects on biofilm formation in end-point assays (*Karatan et al., 2005*; *McGinnis et al., 2009*). We wondered if and how these two polyamines affect each stage of the biofilm lifecycle – biofilm formation *and* biofilm dispersal. To investigate their roles, we assayed the biofilm lifecycle in WT *V. cholerae* and our mutants following exogenous administration of norspermidine and spermidine, alone and in combination. Addition of 100 μM norspermidine strongly promoted biofilm formation and completely prevented biofilm dispersal while 100 μM spermidine dramatically reduced biofilm formation and promoted premature biofilm dispersal (*Figure 3A*, *Video 1*). *vpsL-lux* expression increased by >10-fold following norspermidine treatment and decreased >10-fold following spermidine treatment (*Figure 3B*). These results show that norspermidine and spermidine have opposing activities with regard to biofilm formation and dispersal: norspermidine drives biofilm formation by inducing matrix production, and this prevents biofilm dispersal and spermidine, by suppressing matrix production prevents biofilm formation and drives biofilm dispersal.

We wondered by what mechanism the information encoded in exogenous polyamines was translated into alterations in matrix gene expression and subsequent changes in biofilm dispersal. We reasoned that MbaA and/or PotD1-mediated changes in cytoplasmic c-di-GMP levels could be responsible. To test this possibility, we assessed how changes in cytoplasmic c-di-GMP levels tracked with changes in extracellular polyamine levels by measuring the c-di-GMP reporter output in response to supplied mixtures of norspermidine and spermidine (*Figure 3C*). In the heatmaps, teal and purple represent the lowest and highest values of reporter output, respectively. When provided alone and above a concentration of 10 μM, norspermidine and spermidine strongly increased and decreased, respectively, c-di-GMP reporter activity. At the limit, relative to the WT, norspermidine increased the c-di-GMP reporter output by ~60% while spermidine reduced reporter output by ~25%. Notably, consistent with previous end-point biofilm assays, when norspermidine was present above 50 μM, it overrode the effect of exogenous addition of spermidine, irrespective of spermidine concentration (*Figure 3C*; *McGinnis et al., 2009*). Thus, *V. cholerae* cytoplasmic c-di-GMP levels are responsive to extracellular blends of polyamines. When norspermidine is abundant, *V. cholerae* produces high levels of c-di-GMP, matrix expression is increased, and biofilms do not disperse. In contrast, when spermidine is abundant and norspermidine is absent or present below a threshold concentration, *V. cholerae* c-di-GMP levels drop, matrix production is repressed, and biofilm dispersal occurs.

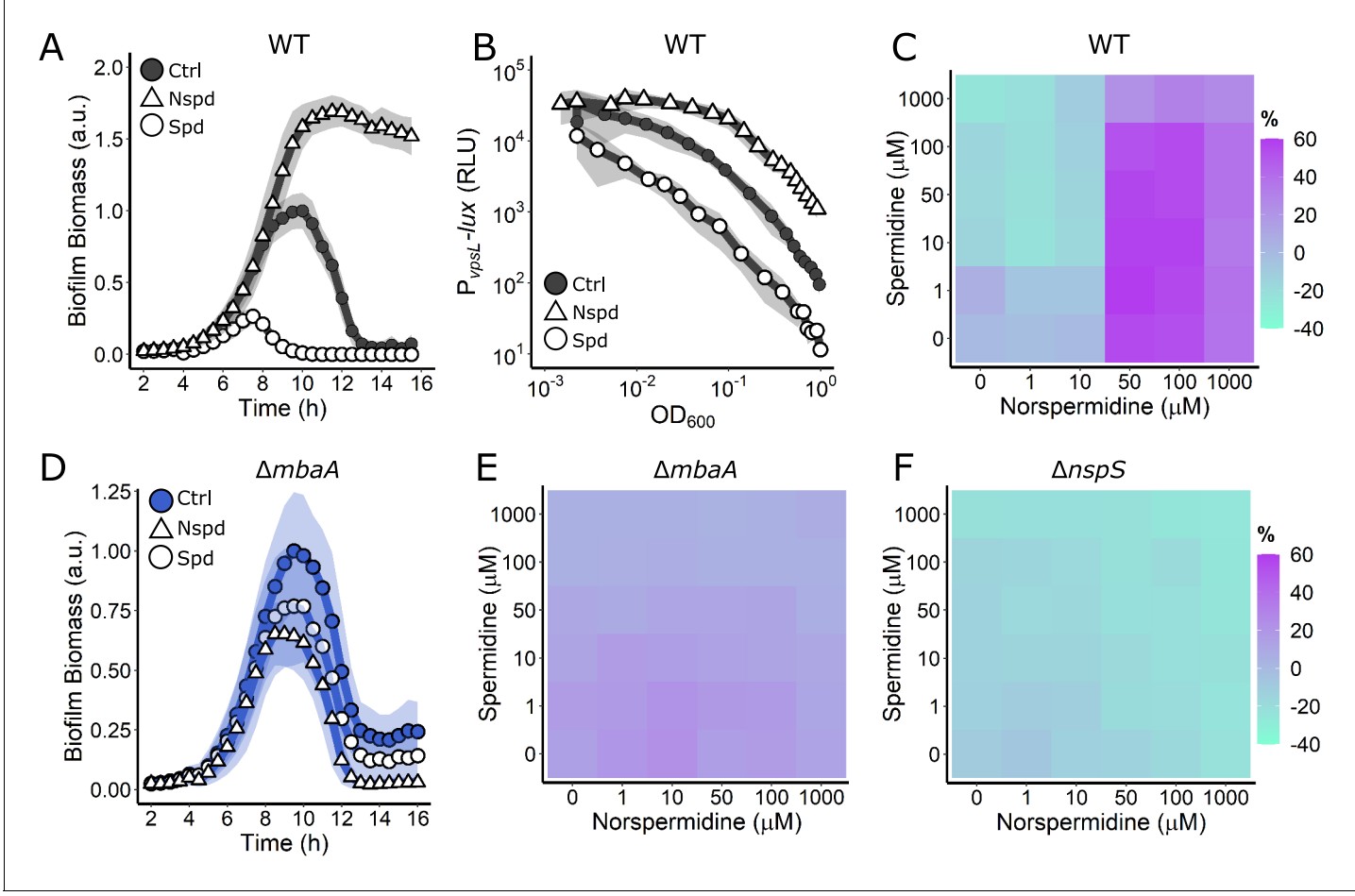

**Figure 3.** Periplasmic detection of polyamines controls *V. cholerae* biofilm dispersal. (A) Quantitation of biofilm biomass over time measured by time-lapse microscopy following addition of water (Ctrl), 100 μM norspermidine, or 100 μM spermidine to WT *V. cholerae*. (B) Light output from the *PvpsL-lux* reporter for the treatments in (A) over the growth curve. (C) c-di-GMP reporter output at the indicated polyamine concentrations for WT *V. cholerae*. Relative reporter signal (% difference) is displayed as a heatmap (teal and purple represent the lowest and highest reporter output, respectively). (D) As in (A) for the Δ*mbaA* mutant. (E) As in (C) for the Δ*mbaA* mutant. (F) As in (C) for the Δ*nspS* mutant. Biofilm biomass data are represented as means normalized to the peak biofilm biomass of the Ctrl condition. In all biofilm biomass measurements, $N = 3$ biological and $N = 3$ technical replicates, ± SD (shaded). a.u.: arbitrary unit. In *vpsL-lux* measurements, $N = 3$ biological replicates, ± SD (shaded). RLU: relative light units. For the c-di-GMP reporter assays, values are expressed as the percentage difference relative to the untreated WT strain, allowing comparisons to be made across all heatmaps in all figures in this article. The same color bar applies to all heatmaps in this article. For each condition, $N = 3$ biological replicates. Numerical values and associated SDs are available in *Supplementary file 1*.

The online version of this article includes the following figure supplement(s) for figure 3:

**Figure supplement 1.** NspS is required for *V. cholerae* biofilm formation.

## MbaA transduces external polyamine information internally to control biofilm dispersal

Our data show that both the MbaA periplasmic polyamine sensor and the PotD1 polyamine importer are required for biofilm dispersal. Our next goal was to distinguish the contribution of periplasmic detection from cytoplasmic import of polyamines to the biofilm lifecycle. We began with periplasmic polyamine detection, mediated by NspS together with MbaA. We supplied norspermidine or spermidine to the Δ*mbaA* mutant and monitored biofilm biomass over time. To our surprise, the Δ*mbaA* mutant was impervious to the addition of either polyamine as both polyamines caused only a very modest reduction in overall biofilm biomass, and dispersal timing resembled the untreated Δ*mbaA* control (*Figure 3D*, *Video 1*). Consistent with this result, titration of the polyamines alone and in combination onto the Δ*mbaA* strain carrying the c-di-GMP reporter did not substantially alter

reporter output (*Figure 3E*). These results show that the dynamic response of WT to polyamines requires MbaA. We suspect that the minor reduction in biofilm production that occurred in the Δ*mbaA* mutant when supplied polyamines is due to non-specific effects, perhaps via interaction of polyamines with negatively charged matrix components. We next investigated the role of the polyamine periplasmic binding protein NspS that transmits polyamine information to MbaA. In the absence of its partner polyamine binding protein, NspS, MbaA is thought to function as a constitutive phosphodiesterase (*Cockerell et al., 2014*). Consistent with this model, deletion of *nspS* in an otherwise WT strain reduced overall peak biofilm biomass by 65% and dispersal initiated 4 hr prior to when dispersal begins in the WT (*Figure 3— figure supplement 1*). In the Δ*nspS* mutant, c-di-GMP levels were lower than in the WT as judged by the c-di-GMP reporter (*Figure 3F*), showing that MbaA is locked as a constitutive phosphodiesterase. Exogenous addition of polyamines had no effect on c-di-GMP levels (*Figure 3F*). Together, these findings demonstrate that the WT *V. cholerae* response to polyamines is controlled by the NspS-MbaA polyamine sensing circuit.

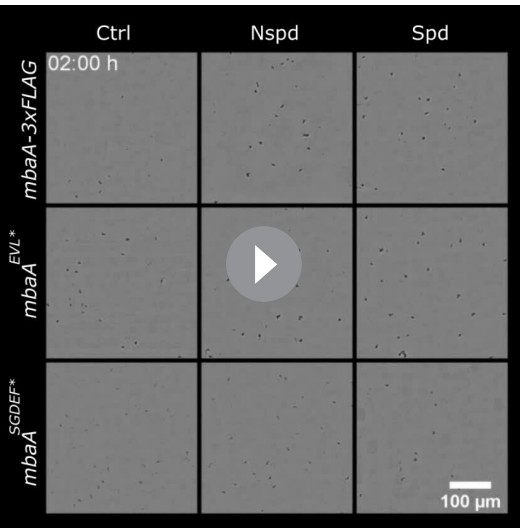

**Video 2.** Representative time-lapse images of the biofilm lifecycles of the *mbaA-3xFLAG*, *mbaA*^EVL*^−*3xFLAG*, and *mbaA*^SGDEF*^−*3xFLAG V. cholerae* strains following treatment with water (Ctrl), 100 μM norspermidine (Nspd), or 100 μM spermidine (Spd).
https://elifesciences.org/articles/65487#video2

## Both the MbaA EVL and SGDEF domains are required for MbaA to detect polyamines

We wondered how the putative MbaA phosphodiesterase and diguanylate cyclase enzymatic activities contribute to the *V. cholerae* responses to norspermidine and spermidine. The cytoplasmic domain of MbaA exhibits phosphodiesterase but not diguanylate cyclase activity in vitro (*Cockerell et al., 2014*). We reasoned that, when an intact regulatory domain is present, MbaA could possess both enzymatic activities, with the activity of each domain inversely controlled by NspS binding. To probe the role of each domain, we introduced inactivating point mutations in the catalytic motifs. To ensure that our mutations did not destabilize MbaA, we first fused MbaA to 3xFLAG and introduced the gene encoding the fusion onto the chromosome at the *mbaA* locus. Tagging did not alter MbaA control over the biofilm lifecycle (*Figure 4—figure supplement 1A*, *Video 2*). To inactivate the MbaA phosphodiesterase activity, we substituted the conserved catalytic residue E553 with alanine, converting EVL to AVL (referred to as EVL* henceforth). This change did not alter MbaA-3xFLAG abundance (*Figure 4—figure supplement 1B*). *V. cholerae* harboring MbaA^EVL*^ exhibited an increase in biofilm biomass and a strong biofilm dispersal defect (*Figure 4A*), and only a modest response to exogenous polyamines, with norspermidine eliciting some inhibition of biofilm dispersal and spermidine driving a small reduction in overall biofilm biomass (*Figure 4B*, *Video 2*). Treatment with norspermidine did increase c-di-GMP levels in *V. cholerae* carrying MbaA^EVL*^ as judged by the reporter output (*Figure 4C*), albeit not to the level of WT (*Figure 3C*), suggesting that the *V. cholerae mbaA*^EVL*^ mutant, which is incapable of c-di-GMP degradation, retains the capacity to synthesize some c-di-GMP via the MbaA SGDEF domain. In contrast, the *V. cholerae mbaA*^EVL*^ strain displayed little reduction in c-di-GMP levels in response to spermidine, presumably because it lacks the phosphodiesterase activity required to degrade c-di-GMP (*Figure 4C*). Thus, despite the fact that the purified cytoplasmic domain functioned only as a phosphodiesterase in vitro, our results suggest that, in vivo, MbaA is capable of synthesizing c-di-GMP in the presence of norspermidine. To validate this prediction, we altered the conserved catalytic residues D426 and E427 to alanine residues, yielding the inactive MbaA SGAAF variant (referred to as SGDEF*). These substitutions did not affect protein levels (*Figure 4—figure supplement 1B*). In

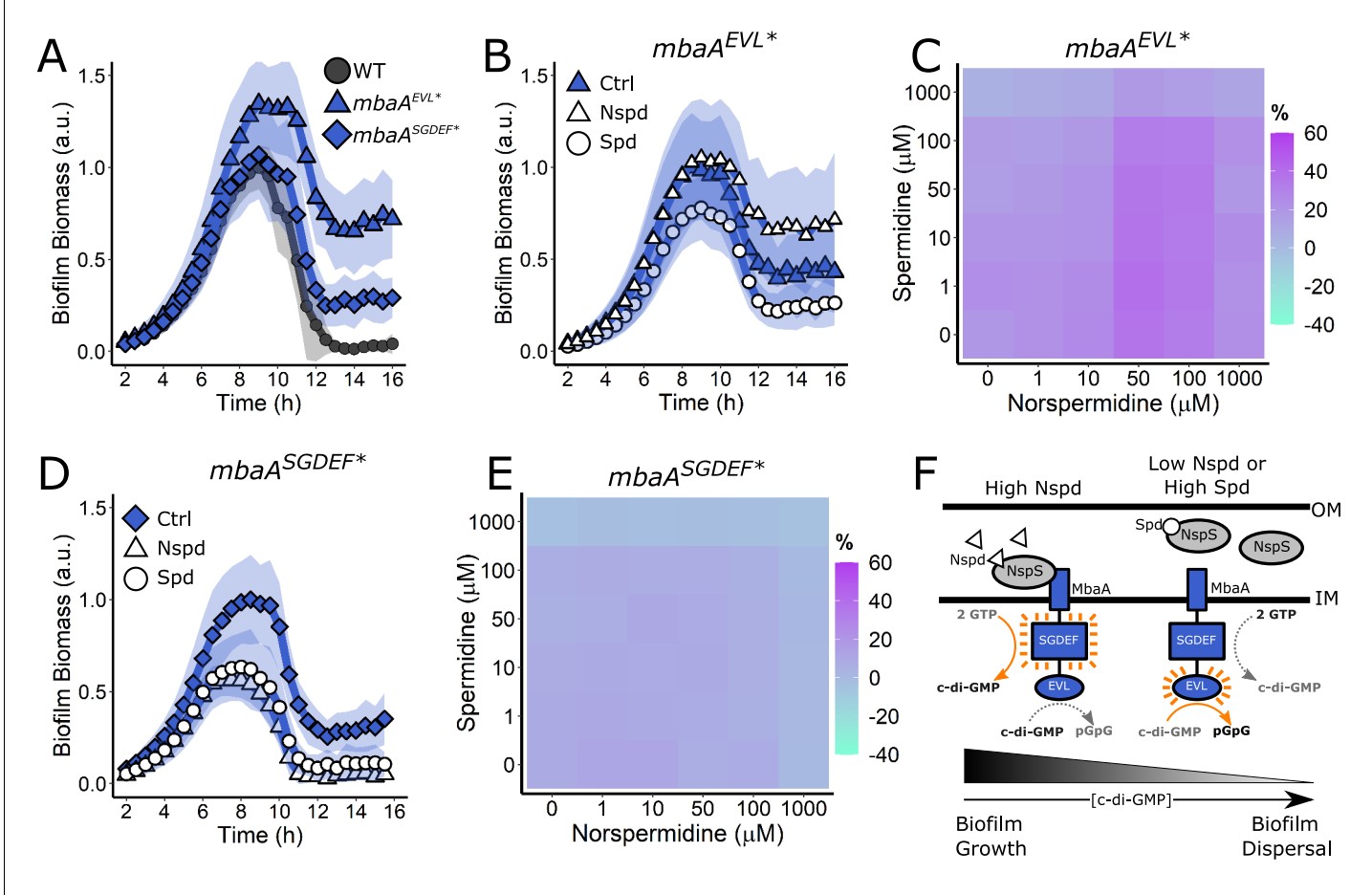

**Figure 4.** Both the MbaA SGDEF and EVL domains are required for regulation of *V. cholerae* biofilm dispersal. (**A**) Quantitation of biofilm biomass over time for the *V. cholerae* strains carrying *mbaA-3xFLAG*, *mbaA*<sup>EVL*</sup>*−3xFLAG*, and *mbaA*<sup>SGDEF*</sup>*−3xFLAG*. (**B**) Quantitation of biofilm biomass over time measured by time-lapse microscopy for *V. cholerae* carrying *mbaA*<sup>EVL*</sup>*−3xFLAG* following addition of water (Ctrl), 100 μM norspermidine, or 100 μM spermidine. (**C**) c-di-GMP reporter output at the indicated polyamine concentrations for *V. cholerae* carrying *mbaA*<sup>EVL*</sup>*−3xFLAG*. Relative reporter signal (% difference) is displayed as a heatmap (teal and purple represent the lowest and highest reporter output, respectively). (**D**) As in (**B**) for the *mbaA*<sup>SGDEF*</sup>*−3xFLAG* mutant. (**E**) As in (**C**) for the *mbaA*<sup>SGDEF*</sup>*−3xFLAG* mutant. (**F**) Schematic representing the proposed MbaA activities in response to norspermidine and spermidine. Biofilm biomass data are represented as means normalized to the peak biofilm biomass of the WT strain or Ctrl condition. In all cases, *N* = 3 biological and *N* = 3 technical replicates, ± SD (shaded). a.u.: arbitrary unit. In the c-di-GMP reporter assays, values are expressed as the percentage difference relative to the untreated WT strain, allowing comparisons to be made across all heatmaps in all figures in this article. The same color bar applies to all panels in this article. For each condition, *N* = 3 biological replicates. Numerical values and associated SDs are available in **Supplementary file 1**. OM: outer membrane; IM: inner membrane.

The online version of this article includes the following figure supplement(s) for figure 4:

**Figure supplement 1.** MbaA active site mutations do not alter protein levels.

every regard, the *mbaA*<sup>SGDEF*</sup> mutant behaved identically to the Δ*mbaA* mutant. The *V. cholerae mbaA*<sup>SGDEF*</sup> mutant exhibited a modest biofilm dispersal defect (**Figure 4A**) and biofilm biomass was reduced in response to exogenous norspermidine and spermidine (**Figure 4D**, **Video 2**). Furthermore, addition of polyamines to the *mbaA*<sup>SGDEF*</sup> mutant harboring the c-di-GMP reporter did not drive substantial changes in reporter output (**Figure 4E**). These results show that the MbaA SGDEF domain is indispensable for the MbaA response to polyamines.

Based on the results in **Figures 3** and **4**, we conclude that the MbaA phosphodiesterase and diguanylate cyclase domains are both required for MbaA to respond properly to polyamines to regulate *V. cholerae* biofilm dispersal. We propose a model in which elevated periplasmic norspermidine levels drive NspS to bind to MbaA. Consequently, MbaA phosphodiesterase activity is suppressed and the diguanylate cyclase activity dominates, which leads to c-di-GMP accumulation

and commitment to the biofilm lifestyle (*Figure 4F*). In contrast, when spermidine is detected, or when periplasmic polyamine concentrations are low, NspS dissociates from MbaA. Consequently, MbaA diguanylate cyclase activity is suppressed and its phosphodiesterase activity dominates, which leads to a reduction in cytoplasmic c-di-GMP and favors biofilm dispersal (*Figure 4F*).

## Polyamines control the *V. cholerae* biofilm lifecycle via periplasmic detection, not via import into the cytoplasm

Here, we investigate the role of PotD1 in controlling biofilm dispersal. Addition of norspermidine did not alter the non-dispersing phenotype of the Δ*potD1* mutant, whereas spermidine treatment drove premature biofilm dispersal and a reduction in peak biofilm biomass (*Figure 5A*, *Video 1*). Consistent with these findings, addition of norspermidine to the Δ*potD1* strain did not alter the c-di-GMP reporter output, whereas addition of spermidine reduced output signal, but only at the highest concentrations tested (*Figure 5B*). These biofilm lifecycle and reporter results show that PotD1-mediated import is not required for spermidine to modulate the biofilm lifecycle. Thus, we considered an alternative mechanism to underlie the *V. cholerae* Δ*potD1* phenotype. We hypothesized that endogenously produced norspermidine is secreted into the periplasm. In the case of WT *V. cholerae*, the equilibrium between norspermidine release and PotD1-mediated reuptake places MbaA into a partially liganded state, and as a consequence, MbaA exhibits modest net phosphodiesterase activity (*Figure 5C*). In the *V. cholerae* Δ*potD1* mutant that cannot reimport norspermidine, we predict that periplasmic norspermidine levels become elevated, NspS detects norspermidine, binds to MbaA, and promotes the MbaA diguanylate cyclase active state (*Figure 5D*). Hence, c-di-GMP is synthesized and biofilm dispersal is prevented. This hypothesis is consistent with our result showing that addition of norspermidine to the Δ*potD1* mutant does not cause any change in biofilm dispersal. Presumably, in the Δ*potD1* mutant, the NspS protein is already saturated due to elevated periplasmic levels of endogenously produced norspermidine. If this hypothesis is correct, norspermidine biosynthesis would be a requirement for biofilm formation because in the absence of norspermidine production, NspS would remain unliganded and MbaA would function as a constitutive phosphodiesterase (*Figure 5E*). Indeed, and consistent with previous results, the Δ*nspC* mutant that lacks the carboxynorspermidine decarboxylase enzyme NspC that is required for norspermidine synthesis failed to form biofilms (*Figure 5F*; *Lee et al., 2009*; *McGinnis et al., 2009*). Furthermore, the Δ*nspC* Δ*potD1* double mutant also failed to form biofilms, showing that endogenous norspermidine biosynthesis is required for the Δ*potD1* mutation to exert its effect (*Figure 5F*). Consistent with these interpretations, the Δ*mbaA* mutation is epistatic to the Δ*potD1* mutation as the Δ*mbaA* Δ*potD1* double mutant exhibited a dispersal phenotype indistinguishable from the single Δ*mbaA* mutant (*Figure 5G*). Administration of exogenous norspermidine or spermidine to the Δ*mbaA* Δ*potD1* double mutant (*Figure 5H*, *Video 1*) mimicked what occurred following addition to the single Δ*mbaA* mutant (*Figure 3D*): that is, essentially no response. Thus, information from externally supplied polyamines is transduced internally by MbaA and not via PotD1-mediated import. Lastly, to confirm that the observed Δ*potD1* phenotype was due to a defect in norspermidine transport, rather than a defect in periplasmic binding and sequestration of norspermidine, we deleted *potA*, encoding the ATPase that supplies the energy required for polyamine import through the PotABCD1 transporter (*Kashiwagi et al., 2002*). In the absence of PotA, PotD1 remains capable of binding polyamines, but no transport occurs. The Δ*potA* mutant exhibited the identical dispersal defect as the Δ*potD1* mutant, demonstrating that transport of norspermidine, not sequestration by PotD1, is the key activity required for WT *V. cholerae* biofilm dispersal (*Figure 5—figure supplement 1*).

To investigate the supposition that the Δ*potD1* strain possesses higher levels of periplasmic norspermidine than WT *V. cholerae*, one needs to isolate and measure polyamines in the periplasmic compartment. Unfortunately, we were unable to reliably obtain periplasmic compartment material. We reasoned, however, that some periplasmic norspermidine would diffuse into the extracellular environment, where it could be isolated and quantified, providing us a means to evaluate differences in periplasmic norspermidine levels between the WT and the Δ*potD1* mutant. At high cell density, approximately 25-fold more norspermidine was present in cell-free culture fluids collected from the Δ*potD1* mutant (average 2.3 μM) than in those prepared from the WT (average 90 nM) (*Figure 5I*). Norspermidine was nearly undetectable in culture fluids from the Δ*nspC* mutant (*Figure 5I*). There was no difference in norspermidine levels in whole-cell extracts prepared from WT (average 0.6 μmol/g) and the Δ*potD1* mutant (average 0.5 μmol/g) (*Figure 5J*). Thus, the difference we measured

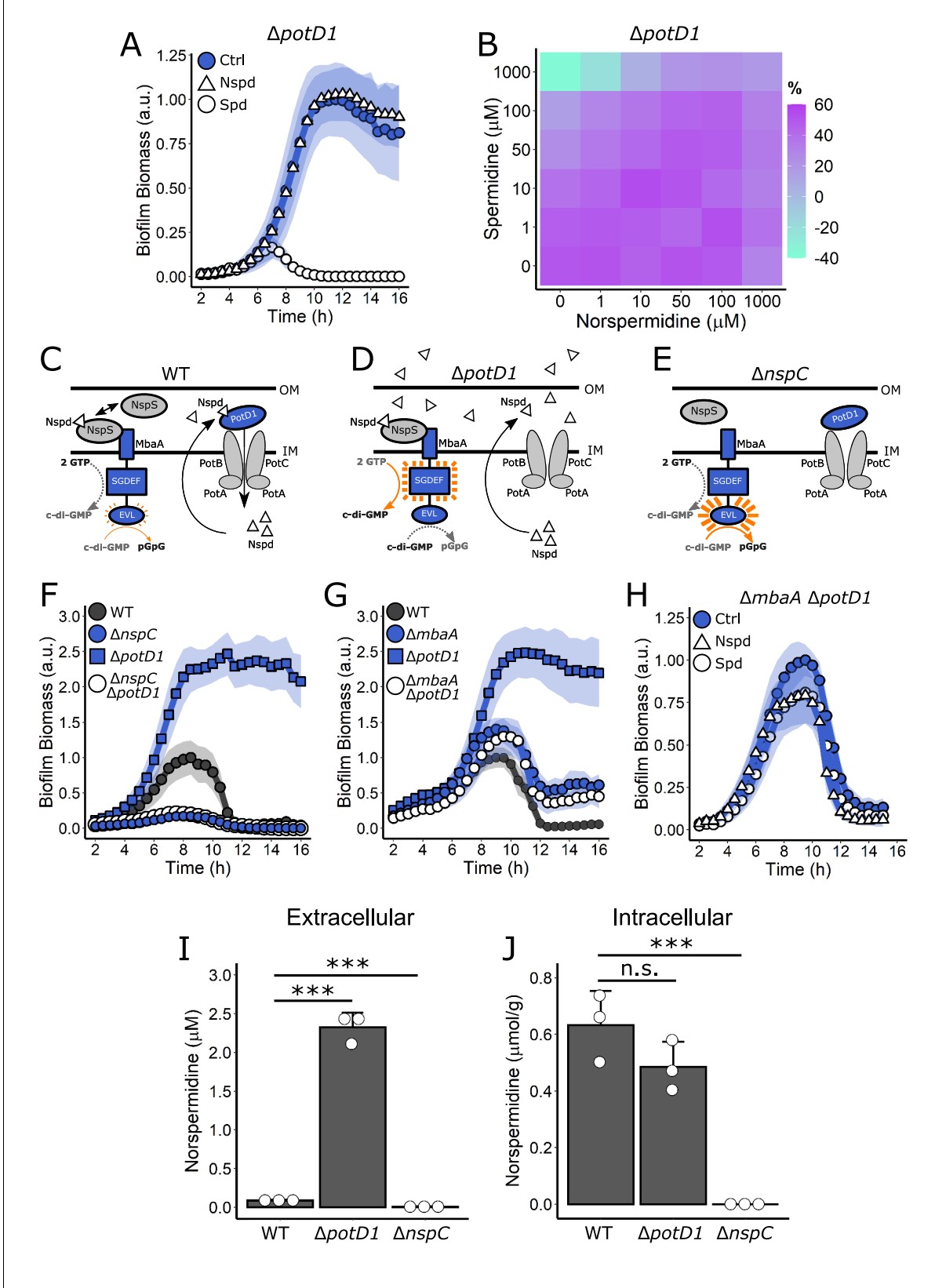

**Figure 5.** Polyamine import is not required for MbaA regulation of *V. cholerae* biofilm dispersal but is required to reduce external norspermidine levels. (**A**) Quantitation of biofilm biomass over time measured by time-lapse microscopy following addition of water (Ctrl), 100 μM norspermidine, or 100 μM

*Figure 5 continued on next page*

Figure 5 continued

spermidine to the ΔpotD1 mutant. (B) c-di-GMP reporter output at the indicated polyamine concentrations for the ΔpotD1 strain. Relative reporter signal (% difference) is displayed as a heatmap (teal and purple represent the lowest and highest reporter output, respectively). Values are expressed as the percentage difference relative to the untreated WT strain, allowing comparisons to be made across all heatmaps in all figures in this article. The same color bar applies to all c-di-GMP reporter heatmaps in this article and for each condition, $N$ = 3 biological replicates. Numerical values and associated SDs are available in *Supplementary file 1*. (C) Schematic of NspS-MbaA periplasmic detection of polyamines and polyamine import by PotD1 in WT *V. cholerae*. OM: outer membrane; IM: inner membrane. (D) Schematic of NspS-MbaA periplasmic detection of norspermidine together with the accumulation of elevated extracellular norspermidine in the *V. cholerae* ΔpotD1 mutant. (E) Schematic of NspS-MbaA activity in the ΔnspC mutant that is incapable of norspermidine biosynthesis. (F) Biofilm biomass over time for WT, the ΔnspC mutant, the ΔpotD1 mutant, and the ΔnspC ΔpotD1 double mutant. (G) As in (F) for WT, the ΔmbaA mutant, the ΔpotD1 mutant, and the ΔmbaA ΔpotD1 double mutant. (H) As in (A) for the ΔmbaA ΔpotD1 double mutant. All biofilm biomass data are represented as means normalized to the peak biofilm biomass of the WT strain or Ctrl condition, and in all cases, $N$ = 3 biological and $N$ = 3 technical replicates, ± SD (shaded). a.u.: arbitrary unit. (I) Concentration of norspermidine in cell-free culture fluids for the WT, ΔpotD1, and ΔnspC strains grown to $OD_{600}$ = 2.0, measured by mass spectrometry. (J) Intracellular norspermidine levels normalized to wet cell pellet mass for the strains in (I). The strains used in (I) and (J) also contained a ΔvpsL mutation to abolish biofilm formation. Thus, all strains existed in the same growth state, which enabled comparisons between strains and, moreover, eliminated any possibility of polyamine sequestration by the biofilm matrix. For (I) and (J), $N$ = 3 biological replicates, error bars represent SDs, and unpaired t-tests were performed for statistical analysis. n.s.: not significant; ***$p<0.001$.

The online version of this article includes the following figure supplement(s) for figure 5:

**Figure supplement 1.** PotA is required for *V. cholerae* biofilm dispersal.

**Figure supplement 2.** Polyamine levels for WT, ΔpotD1, and ΔnspC strains.

---

in norspermidine concentrations in the two extracellular fractions cannot be due to variations in norspermidine biosynthesis between the two strains. As expected, both the cell-free fluid and the whole-cell extract from the ΔnspC mutant were devoid of norspermidine and spermidine (*Figure 5J*, *Figure 5—figure supplement 2*), and consistent with previous reports (*Lee et al., 2009*), only trace spermidine could be detected in the WT intra- and extracellular preparations (*Figure 5—figure supplement 2*). For reference, we also measured the intra- and extracellular levels of additional polyamines from the same samples (*Figure 5—figure supplement 2*). Thus, our results argue against a cytoplasmic role for norspermidine in the biofilm lifecycle. Rather, our results indicate that the non-dispersing phenotype of the ΔpotD1 strain is due to elevated accumulation of periplasmic norspermidine stemming from the failure of the mutant to reuptake self-made norspermidine. The consequence is that, compared to WT *V. cholerae*, in the ΔpotD1 mutant, MbaA is biased in favor of c-di-GMP production and commitment to the biofilm lifestyle (*Figure 5D*).

## MbaA and polyamine levels remain constant throughout the *V. cholerae* biofilm lifecycle

Our results demonstrate that WT *V. cholerae* is poised to respond to exogenous polyamines through the NspS-MbaA signaling circuit, presumably allowing *V. cholerae* to adapt to the species composition of its environment. We wondered if, under the conditions of our biofilm assay, MbaA receptor abundance or endogenously produced norspermidine/spermidine levels change over the course of the biofilm lifecycle, either or both of which could influence the normal biofilm growth to dispersal transition. To test these possibilities, we measured MbaA-3xFLAG protein levels and the concentrations of intra- and extracellular polyamines during biofilm formation (T = 5 hr) and after dispersal (T = 10 hr) in WT *V. cholerae*. MbaA abundance and the intracellular concentrations of norspermidine and spermidine did not fluctuate between the 5 hr and 10 hr timepoints (*Figure 6A, B*). The concentration of extracellular norspermidine did increase between 5 hr and 10 hr, from <10 nM to ~75 nM (*Figure 6B*); however, this range is far below the NspS-MbaA detection threshold (*Figure 3C*). Extracellular spermidine was nearly undetectable at both timepoints (*Figure 6B*). Measurements of additional polyamines from the same samples are shown in *Figure 6—figure supplement 1*. We take these results to mean that, in the absence of exogenously supplied polyamines, MbaA activity is constant and, in our experiments, the enzyme is modestly biased toward phosphodiesterase function throughout the biofilm lifecycle (as indicated by the ΔmbaA strain phenotypes shown in *Figure 2B, C*).

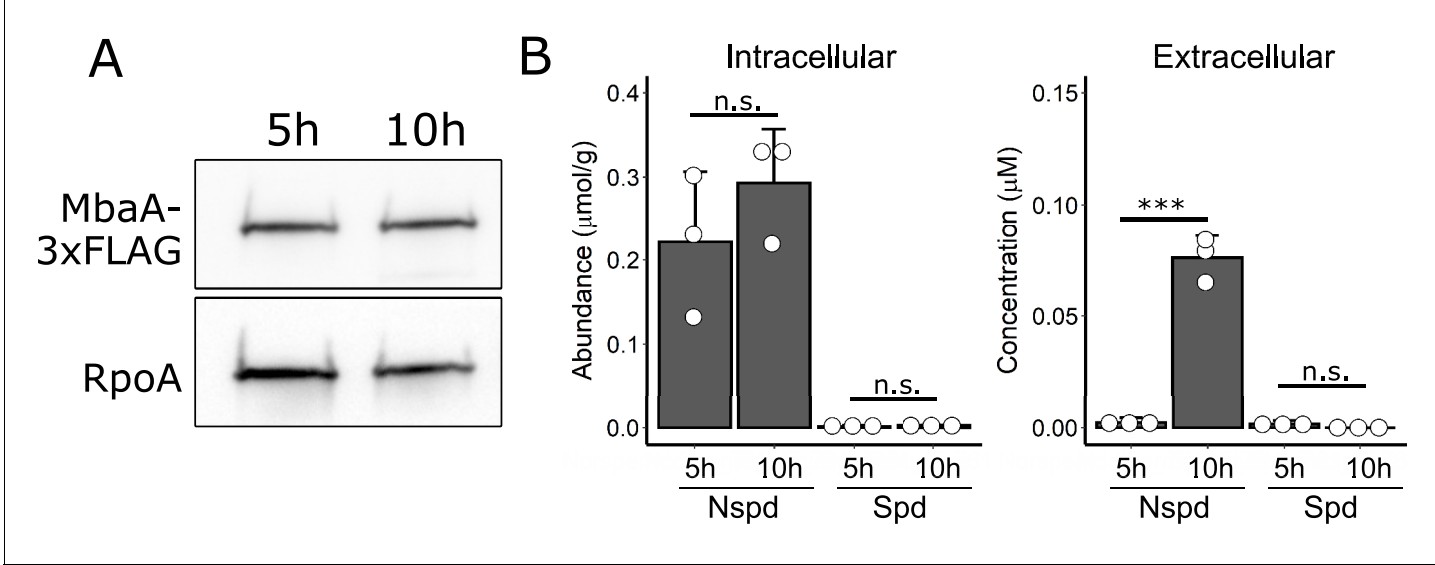

**Figure 6.** MbaA and intracellular polyamine levels remain constant throughout the biofilm lifecycle. (**A**) Top panel: western blot showing MbaA-3xFLAG levels at 5 hr and 10 hr post inoculation. Bottom panel: the RpoA loading control. Data are representative of three biological replicates. (**B**) Measurements of intracellular (left panel) and extracellular (right panel) norspermidine (Nspd) and spermidine (Spd) levels at 5 hr and 10 hr post inoculation. Intracellular levels were normalized to wet cell pellet mass. $N$ = 3 biological replicates, error bars represent SDs, and unpaired t-tests were performed for statistical analysis. n.s.: not significant; ***p<0.001.

The online version of this article includes the following figure supplement(s) for figure 6:

**Figure supplement 1.** Polyamine levels in WT *V. cholerae* prior to and following biofilm dispersal.

## Discussion

In this study, we investigated the effects of the polyamines norspermidine and spermidine on *V. cholerae* biofilm dispersal. Norspermidine and spermidine, which differ in structure only by one methylene group, mediate starkly opposing effects on the *V. cholerae* biofilm lifecycle: norspermidine inhibits and spermidine promotes biofilm dispersal. Both function through the NspS-MbaA circuit. Thus, the polyamine binding protein NspS must harbor the exquisite capability to assess the presence or absence of a single chemical moiety as norspermidine binding drives NspS interaction with MbaA, whereas binding to spermidine prevents this interaction. Here, we speculate on the possible biological significance of our findings. We suspect that norspermidine and spermidine act as 'self' and 'other' cues, respectively. Specifically, norspermidine is a rare polyamine in the biosphere, produced only by select organisms, namely *V. cholerae* and closely related marine vibrios (*Hamana, 1997*; *Michael, 2018*; *Yamamoto et al., 1991*). The observation that laboratory-grown WT *V. cholerae* releases little norspermidine, at least in part due to PotD1-mediated reuptake (*Figure 5C*), suggests that this system does not behave like a canonical quorum-sensing pathway. However, it is possible that norspermidine is secreted by *V. cholerae* under some environmental conditions, or by other vibrios, and *V. cholerae* detects the released norspermidine via NspS-MbaA, and its dispersal from biofilms is prevented. Thus, when close relatives are nearby, as judged by detection of the presence of norspermidine, *V. cholerae* elects to remain in its current biofilm niche. More speculative is the notion that norspermidine functions as a cue for phage-, predator-, or toxin-induced cell lysis. Specifically, if *V. cholerae* can detect norspermidine released by neighboring lysed *V. cholerae* cells, in response to the perceived danger, *V. cholerae* would remain in the protective biofilm state. Conversely, detection of spermidine, a nearly ubiquitously produced polyamine (*Michael, 2018*), could alert *V. cholerae* to the presence of competing or unrelated organisms. In this case, *V. cholerae* would respond by dispersing from biofilms and fleeing that locale. In a seemingly parallel scenario, we previously discovered that autoinducer AI-2, a universally produced quorum-sensing signal, also drove *V. cholerae* biofilm repression and premature dispersal, while CAI-1, the *V. cholerae* 'kin' quorum-sensing autoinducer, did not (*Bridges and Bassler, 2019*). Together,

our findings suggest that *V. cholerae* possesses two mechanisms to monitor its own numbers (norspermidine and CAI-1) and two mechanisms to take a census of unrelated organisms in the environment (spermidine and AI-2), and it responds by remaining in the biofilm state when closely related species are in the majority and by exiting the biofilm state when the density of non-related organisms is high, presumably to escape competition/danger.

We found that periplasmic polyamine detection by NspS-MbaA, but not polyamine import, controls the biofilm dispersal program and both the MbaA EVL and SGDEF domains are required for the *V. cholerae* response to polyamines. Despite previous work suggesting that MbaA functions exclusively as a phosphodiesterase and that its SGDEF domain, which possesses a serine substitution in the active site relative to the canonical GGDEF active site, is inert with respect to c-di-GMP synthesis, our data suggest that when norspermidine is present, MbaA does synthesize c-di-GMP (*Figure 3C*, *Figure 4C*), as has also been shown for other SGDEF proteins (*Pérez-Mendoza et al., 2011*). Therefore, MbaA is likely a bifunctional protein capable of both c-di-GMP synthesis and degradation. We wonder how such an arrangement benefits *V. cholerae* in its response to polyamines. We speculate that the ability of MbaA to inversely alter c-di-GMP levels in response to two discrete cues coordinates polyamine signaling and enables more rapid and greater magnitude changes in c-di-GMP levels than would be achievable if two separate receptors existed, each harboring only a single catalytic activity and each responsive to only one polyamine.

An analogous bifunctional arrangement exists for the quorum-sensing two-component receptor LuxN from *Vibrio harveyi*. LuxN is an inner membrane receptor that harbors a periplasmic sensory domain responsible for detection of a self-made small molecule cue, a homoserine lactone agonist with a four-carbon acyl tail (*Bassler et al., 1994*). The ligand encodes species-specific information about cell population density. Competing bacterial species that co-occupy the niche in which *V. harveyi* resides produce similar acyl homoserine lactone signal molecules, and notably, those molecules differ only in the acyl tail length or decoration (*Papenfort and Bassler, 2016*). The tails of the molecules produced by the competitors usually possess more than four carbons, and they act as potent LuxN antagonists. With respect to mechanism, the self-made agonist drives LuxN into phosphatase mode while the antagonists induce LuxN to act as a kinase (*Ke et al., 2015*). Thus, as in the MbaA circuit, binding of the different ligands to LuxN propels opposing enzymatic activities in the cytoplasm. We speculate that ligand-driven alternations between opposing enzymatic activities could be an underappreciated mechanism that sensors employ to transmit species-specific and species-non-specific information into discrete changes in downstream behaviors.

In addition to conveying information about the species composition of the vicinal community, the MbaA SGDEF and EVL domains could also participate in dimerization and/or allosteric binding to c-di-GMP or other metabolites to mediate feedback regulation of the opposing MbaA activities. Parallel examples exist, here we present one: *Caulobacter crescentus* contains a GGDEF/EAL protein called CC3396, in which the GGDEF domain is non-catalytic but it nonetheless binds to GTP, which allosterically activates the phosphodiesterase activity in the neighboring EAL domain (*Christen et al., 2005*). A feedback mechanism could provide MbaA with the ability to integrate the information encoded in extracellular polyamine blends with cytoplasmic cues, such as the metabolic state of the cell or the current cytoplasmic c-di-GMP level. Taken together, the double ligand detection capability linked to the dual catalytic activities of the NspS-MbaA circuit allows *V. cholerae* to distinguish between remarkably similar ligands and, in response, have the versatility to convey distinct information into the cell to alter the biofilm lifecycle.

In summary, four signaling pathways have now been defined that feed into the regulation of *V. cholerae* biofilm dispersal: starvation (RpoS) (*Singh et al., 2017*), quorum sensing (via CAI-1 and AI-2) (*Bridges and Bassler, 2019*; *Singh et al., 2017*), the recently identified DbfS/DbfR cascade (ligand unknown) (*Bridges et al., 2020*), and through the current work, polyamine signaling via NspS-MbaA. We propose that integrating the information contained in these different stimuli into the control of biofilm dispersal endows *V. cholerae* with the ability to successfully evaluate multiple features of its fluctuating environment prior to committing to the launch of this key lifestyle transition that, ultimately, impinges on its overall survival. Because *V. cholerae* biofilm formation and biofilm dispersal are intimately connected to cholera disease and its transmission, we suggest that deliberately controlling the biofilm lifecycle, possibly via synthetic strategies that target polyamine signal transduction via NspS-MbaA, could be a viable therapeutic strategy to ameliorate disease.

## Materials and methods

### Bacterial strains, reagents, and imaging assays

The *V. cholerae* strain used in this study was WT O1 El Tor biotype C6706str2. Antibiotics were used at the following concentrations: polymyxin B, 50 µg/mL; kanamycin, 50 µg/mL; spectinomycin, 200 µg/mL; chloramphenicol, 1 µg/mL; and gentamicin, 15 µg/mL. Strains were propagated for cloning purposes in lysogeny broth (LB) supplemented with 1.5% agar or in liquid LB with shaking at 30°C. For biofilm dispersal analyses, *lux* measurements, c-di-GMP reporter quantitation, and norspermidine measurements, *V. cholerae* strains were grown in M9 medium supplemented with 0.5% dextrose and 0.5% casamino acids. All strains used in this work are reported in the Key resources table. Compounds were added from the onset of biofilm initiation. Norspermidine (Millipore Sigma, I1006-100G-A) and spermidine (Millipore Sigma, S2626-1G) were added at the final concentrations designated in the figures. The biofilm lifecycle was measured using time-lapse microscopy as described previously (*Bridges and Bassler, 2019*). All plots were generated using ggplot2 in R. Light production driven by the *vpsL* promoter was monitored as described previously (*Bridges et al., 2020*). Results from replicates were averaged and plotted using ggplot2 in R.

### DNA manipulation and strain construction

All strains generated in this work were constructed by replacing genomic DNA with DNA introduced by natural transformation as previously described (*Bridges et al., 2020*). PCR and Sanger sequencing were used to verify correct integration events. Genomic DNA from recombinant strains was used for future co-transformations and as templates for PCR to generate DNA fragments, when necessary. See Key resources table for primers and g-blocks (IDT) used in this study. Gene deletions were constructed in frame and eliminated the entire coding sequences. The exceptions were *mbaA* and *nspS*, which overlap with adjacent genes, so an internal portion of each gene was deleted, ensuring that adjacent genes were not perturbed. To construct *mbaA-3xFLAG* at the native *mbaA* locus, gene synthesis was used to preserve the coding sequence of the downstream gene. To achieve this, the overlapping region was duplicated. In the duplication, the start codon of the downstream gene was disabled by mutation. The coding sequencing of *mbaA* was preserved. The DNA specifying *3xFLAG* was introduced immediately upstream of the *mbaA* stop codon. All strains constructed in this study were verified using the Genewiz sequencing service.

### Western blotting

Cultures of strains carrying MbaA-3xFLAG and relevant catalytic site variants were collected at $OD_{600} = 1.0$ and subjected to centrifugation for 1 min at 13,000 rpm. The pellets were flash frozen, thawed for 5 min at 25°C, and subsequently chemically lysed by resuspending to $OD_{600} = 1.0$ in 75 µL Bug Buster (Novagen, #70584-4) supplemented with 0.5% Triton-X, 50 µg/mL lysozyme, 25 U/mL benzonase nuclease, and 1 mM phenylmethylsulfonyl fluoride for 10 min at 25°C. Lysates were solubilized in $1\times$ SDS-PAGE buffer for 1 hr at 37°C. Samples were loaded into 4–20% Mini-Protein TGX gels (Bio-Rad). Electrophoresis was carried out at 200 V. Proteins were transferred from the gels to PVDF membranes (Bio-Rad) for 50 min at 4°C at 100 V in 25 mM Tris base, 190 mM glycine, 20% methanol. Following transfer, membranes were blocked in 5% milk in PBST (137 mM NaCl, 2.7 mM KCl, 8 mM $Na_2HPO_4$, 2 mM $KH_2PO_4$, and 0.1% Tween) for 1 hr, followed by three washes with PBST. Subsequently, membranes were incubated for 1 hr with a monoclonal Anti-FLAG-Peroxidase antibody (Millipore Sigma, #A8592) at a 1:5000 dilution in PBST with 5% milk. After washing six times with PBST for 5 min each, membranes were exposed using the Amersham ECL western blotting detection reagent (GE Healthcare). For the RpoA loading control, the same protocol was followed except that the primary antibody was Anti-*Escherichia coli* RNA Polymerase α (Biolegend, #663104) used at a 1:10,000 dilution and the secondary antibody was an Anti-Mouse IgG HRP conjugated antibody (Promega, #W4021) used at a 1:10,000 dilution.

### Extraction and measurements of polyamines

To measure intracellular polyamines, previously established techniques were followed with slight modifications (*McGinnis et al., 2009*). For planktonic cell measurements, *V. cholerae* strains were grown in 5 mL of M9 medium containing glucose and casamino acids with constant shaking at 30°C.

2.0 $OD_{600}$ equivalents of *V. cholerae* cells were collected by centrifugation for 1 min at 13,000 rpm from cultures that had been grown to $OD_{600}$ ~2.0. Cells were washed once with 1× PBS and subsequently weighed and resuspended in 100 µL PBS. The resuspended cells were lysed by 10 freeze-thaw cycles in liquid nitrogen. After the final freeze-thaw cycle, samples were subjected to bath sonication for 20 s (Fisher Scientific, FS30) and subsequently diluted to 100 mg/mL in 1× PBS. To precipitate proteins, 100 µL of 50% trichloroacetic acid was added to 500 µL of lysate and the mixtures were incubated for 5 min. Precipitated material was pelleted by centrifugation for 5 min at 13,000 rpm. 500 µL of the resulting clarified supernatants were subjected to benzoylation as described below. For measurements of polyamines in cell-free culture fluids, the culture fluids from the samples described above were collected following the centrifugation step. Remaining cells were removed by filtration with 0.45 µm filters (Millex-HV), proteins were precipitated as above, and the benzoylation derivatization was performed. For measurements of polyamine concentrations in static cultures at the 5 hr and 10 hr timepoints, the procedure was the same as above with the following modifications: 10 mL cultures were incubated at 30℃ without shaking to achieve conditions favoring the biofilm lifecycle. In the case of the 5 hr cultures, samples were briefly mixed by vortex with sterile 1 mm glass beads to dislodge biofilms prior to centrifugation. Cell pellets from both the 5 hr and 10 hr samples were resuspended in 1× PBS at 40 mg/mL, proteins were precipitated, and 500 µL was used for derivatization.

Samples were derivatized with benzoyl chloride as previously described, except that the amount of benzoyl chloride was doubled to ~80 µmol, and benzylated polyamines were resuspended in 200 µL of the mobile phase (*Morgan, 1998*). In all cases, medium blanks and polyamine standards were simultaneously derivatized and analyzed to generate calibration curves. HPLC was performed on a Shimadzu UFLC system with PAL autoinjector. 30 min gradient chromatography separation was performed using solvent A (10% methanol/90% water) and solvent B (95% methanol/5% water) on an ACE Ultracore 2.5 SuperC18 1.0 × 50 mm column with 63 µL/min flow rate at a column temperature of 54℃. Mass spectrometry was performed using an Orbitrap XL mass spectrometer (Thermo) with an APCI ionization source in positive mode. The parent ion (MS1) was detected in the Orbitrap with 100,000 mass resolution, and fragment ions (MS2) were detected in the ion trap. Parameters were as follows: the vaporizer temperature was 270℃, the sheath gas flow rate was 18 a.u., the auxiliary gas flow rate was 5 a.u., the sweep gas flow rate was 5 a.u., the discharge current was 9 mA, the capillary temperature was 250℃, the capillary voltage was 36 V, and the tube lens was 72 V. Skyline software (University of Washington) was used to analyze results.

### c-di-GMP reporter assays

The c-di-GMP reporter has been described (*Zamorano-Sánchez et al., 2019*; *Zhou et al., 2016*). To measure relative reporter output for each condition, 100 µL of *V. cholerae* cultures were back diluted to $OD_{600}$ = 0.0002 following overnight growth. Cultures were dispensed into 96-well plates containing the indicated polyamines, and the plates were covered in breathe-easy membranes to prevent evaporation. Samples were incubated overnight at 30℃ with shaking. The following morning, the breathe-easy membranes were removed and fluorescence measurements were obtained using a Bio-Tek Synergy Neo2 Multi-Mode reader. For AmCyan, the excitation wavelength was 440 ± 20 nm and emission was detected at 490 ± 20 nm; and for TurboRFP, the excitation wavelength was 530 ± 20 nm and the emission was 575 ± 20 nm. The c-di-GMP-regulated TurboRFP fluorescence was divided by the constitutive AmCyan fluorescence to yield the relative fluorescence intensity (RFI). To facilitate comparisons between strains and conditions, RFIs were subsequently normalized to the untreated *V. cholerae* WT RFI and the data are expressed as the percentage differences (denoted relative reporter signal). All results were obtained in biological triplicate, and data analysis and plotting were performed in R.

## Acknowledgements

We thank members of the Bassler group and Prof. Ned Wingreen for thoughtful discussions. The c-di-GMP reporter plasmid was a kind gift from Fitnat Yildiz (UC Santa Cruz). Mass spectrometry was conducted by the Princeton Molecular Biology Proteomics and Mass Spectrometry Core Facility. This work was supported by the Howard Hughes Medical Institute, NIH Grant 5R37GM065859, National Science Foundation Grant MCB-1713731, and a Max Planck-Alexander von Humboldt

research award to BLB. AAB is a Howard Hughes Medical Institute Fellow of the Damon Runyon Cancer Research Foundation, DRG-2302–17. The content is solely the responsibility of the authors and does not necessarily represent the official views of the National Institutes of Health. The funders had no role in study design, data collection and analysis, decision to publish, or preparation of the manuscript.

## Additional information

### Funding

| Funder | Grant reference number | Author |
|---|---|---|
| Howard Hughes Medical Institute | | Bonnie L Bassler |
| National Institutes of Health | 5R37GM065859 | Bonnie L Bassler |
| National Science Foundation | MCB-1713731 | Bonnie L Bassler |
| Max Planck - Alexander von Humboldt-Stiftung | | Bonnie L Bassler |
| Damon Runyon Cancer Research Foundation | DRG-2302-17 | Andrew A Bridges |

The funders had no role in study design, data collection and interpretation, or the decision to submit the work for publication.

### Author contributions

Andrew A Bridges, Conceptualization, Resources, Data curation, Formal analysis, Funding acquisition, Validation, Investigation, Visualization, Methodology, Writing - original draft, Writing - reviewing and editing; Bonnie L Bassler, Conceptualization, Resources, Data curation, Formal analysis, Supervision, Funding acquisition, Validation, Investigation, Writing - original draft, Project administration, Writing - review and editing

### Author ORCIDs

Andrew A Bridges (iD) https://orcid.org/0000-0002-8132-751X
Bonnie L Bassler (iD) https://orcid.org/0000-0002-0043-746X

### Decision letter and Author response

Decision letter https://doi.org/10.7554/eLife.65487.sa1
Author response https://doi.org/10.7554/eLife.65487.sa2

## Additional files

### Supplementary files

- Supplementary file 1. c-di-GMP reporter output averages and standard deviations.

- Transparent reporting form

### Data availability

All data generated and analyzed in this study are included in the manuscript and supporting files. Source data files have been provided in Zenodo (https://doi.org/10.5281/zenodo.4651348).

The following dataset was generated:

| Author(s) | Year | Dataset title | Dataset URL | Database and Identifier |
|---|---|---|---|---|
| Bridges AA, Bassler BL | 2021 | Inverse regulation of *Vibrio cholerae* biofilm dispersal by polyamine signals | https://doi.org/10.5281/zenodo.4651348 | Zenodo, 10.5281/zenodo.4651348 |

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
