## [Decision Letter]

**Acceptance summary:**

This paper presents evidence that the polyamines spermidine and norspermidine can inversely impact the dispersal of *Vibrio cholerae* biofilms through the protein MbaA, which contains a periplasmic domain that may interface with a possibly direct polyamine sensor, NspS. The paper presents compelling data to support the notion that the cyclic-di-GMP synthetase and phosphodiesterase domains of MbaA are ultimately responsible for transducing polyamine signals in the periplasm into changes in biofilm formation. The work indicates that external polyamines may function to provide self/non-self information, similar to quorum sensing systems.

**Decision letter after peer review:**

Thank you for submitting your article "Inverse regulation of *Vibrio cholerae* biofilm dispersal by polyamine signals" for consideration by *eLife*. Your article has been reviewed by 3 peer reviewers, one of whom is a member of our Board of Reviewing Editors, and the evaluation has been overseen by Gisela Storz as the Senior Editor. The following individuals involved in review of your submission have agreed to reveal their identity: Anthony Michael (Reviewer #2); Mark Mandel (Reviewer #3).

The reviewers have discussed the reviews with one another and the Reviewing Editor has drafted this decision to help you prepare a revised submission.

Summary:

This paper presents evidence that the polyamines spermidine and norspermidine can inversely impact the dispersal of *Vibrio cholerae* biofilms through the protein MbaA, which contains a periplasmic domain that may interface with a possibly direct polyamine sensor, NspS. The paper is easy to follow and presents some compelling data to support the notion that the cyclic-di-GMP synthetase and phosphodiesterase domains of MbaA are ultimately responsible for transducing polyamine signals in the periplasm into changes in biofilm formation. The work indicates that external polyamines may function to provide self/non-self information, similar to quorum sensing systems. There were several major concerns raised, as summarized below. In particular, there was concern from all three reviewers about how much polyamine-based signaling contributes to the formation of biofilms by wild-type cells, especially the self-produced norspermidine. In short, there's currently confusion about whether polyamines only impact biofilms when added exogenously or whether the endogenously produced/secreted levels can and do impact biofilms. Measuring these polyamines in biofilm cultures will be essential to a revision.

Essential revisions:

Experimental issues:

The deletion mutants (e.g. in Figure 2) need to be complemented.

The evidence for the periplasmic sensing model in the ΔpotD1 mutant is strong, but could be further strengthened by an intervention that blocks the proposed norspermidine export. If there is a way to interrupt endogenous norspermidine synthesis and/or export to the periplasm, this would add further support for the model in Figure 5CD to explain the role of PotD1. Similarly, a PotD1 allele that is still present in the periplasm but is nonfunctional for import would provide evidence that the key issue is flux/import, and not sequestration for PotD1.

The fact that the mbaA(SGDEF)* mutant still accumulates biomass at the same rate up to 8 hr as the wild-type with the same max value suggests that this pathway doesn't become relevant until later in biofilm formation. But, spermidine at high levels can inhibit early on (Figure 3A). To me, these two findings aren't reconciled in the authors' model (Figure 4F) which would lead one to believe that CDG synthesis by MbaA is critical for biofilm formation, but it's not (see Figure 4A) – to be clear, I recognize that the paper's title and abstract focus on dispersal, but the model leads one to think that the pathway being studied here is important throughout the biofilm development process. Maybe more generally, the issue is about how spermidine and norspermidine impact the normal development of a wild-type biofilm, if at all. It's not revealed until the Discussion that norspermidine isn't likely secreted at appreciable levels (something that should definitely be noted in the Introduction), so while exogenous addition can impact biofilms, the lack of secreted norspermidine makes it sound like it's actually not so critical to our understanding of WT biofilms and the behavior of WT biofilms documented in Figure 2. Following on this point above, I have two questions that seem critical to assessing how polyamines impact biofilm development: (i) What's the level of spermidine produced and secreted? If the dispersal is normally driven by spermidine, its levels should accumulate extracellularly or periplasmically at the time biofilms being to disperse. (ii) What's the biofilm phenotype of a mutant that cannot synthesize spermidine or norspermidine or both? It could be that mutants that can't synthesize polyamines behave like the WT and that neither is detectable in the periplasm/secreted fraction – such results wouldn't negate or diminish the conclusion that these polyamines can impact biofilm formation when added exogenously (which the Discussion speculates on), but it would mean that the usual pattern of biofilm mass accumulation and then drop off noted in Figure 2B isn't driven by the polyamines. At the very least, the initial accumulation seems very unlikely to be driven by norspermidine if it can't be detected.

Following on the point above, the group of Karatan, using HPLC, has repeatedly failed to detect extracellular norspermidine in the spent growth medium of *V. cholerae* planktonic and biofilm cultures (refs. Wotanis et al., 2017 and uncited Parker et al., (2012) FEMS Microbiol Lett 329, 19-27). The current authors have used LC-MS/MS to determine that spent growth medium from planktonic potD1 gene deletion cultures contain 15 times more norspermidine than the parental wildtype strain. The authors need to provide a quantitative analysis of the absolute levels of norspermidine produced by biofilm cultures and assess whether it exceeds the minimum amount required to see effects when exogenously adding norspermidine.

The hypothesis of the authors – that self-produced norspermidine in the Δ-potD1 mutant accumulates in the periplasm and elicits biofilm effects due to its periplasmic concentration, is elegant but requires two suppositions: firstly, norspermidine is secreted by the cell into the periplasm, and that the periplasmic concentration of norspermidine must be above a threshold that results in elicitation of biofilm and dispersal effects. Norspermidine is presumably able to freely diffuse out of the periplasm into the external medium via porins. Given the time scales involved, the concentration of norspermidine would be likely to equilibrate either side of the outer membrane. Therefore, either the concentration of norspermidine in the external cell-free medium must be above a threshold level that elicits biofilm effects, as discussed above. Or, norspermidine must be sequestered in the periplasm. How might sequestration of the norspermidine in the periplasm occur? The Karatan group showed that the cell-free spent medium from V. cholerae biofilm cultures contained 2 mM cadaverine, whereas the sterile cell-free growth medium contained only 3 μM cadaverine. This indicates that the biofilm cells were undergoing acid stress, which induces the cadaverine-producing and exporting cadAB system (lysine decarboxylase and lysine/cadaverine antiporter) but more relevantly, endogenously-produced cadaverine has been shown to block outer membrane porin channels, even if produced under pH neutral conditions (Samartzidou and Delcour (1999) "Excretion of endogenous cadaverine leads to a decrease in porin-mediated outer membrane permeability", J. Bacteriol., 181, 791-798). Production and excretion of cadaverine by CadAB during biofilm formation could prevent self-produced norspermidine diffusing out of the periplasm. In the WT cells, any periplasmic norspermidine could be reacquired by the cell through the potABCD transporter, whereas in the Δ-potD1 mutant, a cadaverine block of porins might result in accumulation of periplasmic norspermidine. Questions to the authors: (1) What was the concentration of cadaverine in the WT and Δ-potD1 cell cultures and cell-free spent medium? (2) What was the concentration of putrescine, spermidine and N-acetylnorspermidine? Presumably this data is part of the original LC-MS/MS output. (3) The authors' polyamine data was obtained with planktonic cultures containing an additional deletion of the vps1 gene that eliminates the biofilm matrix. Did the authors measure norspermidine levels in cell-free spent medium from biofilm cultures? If not, it would be very relevant to the authors' hypothesis to measure norspermidine and cadaverine in the cell-free spent medium of WT and Δ-potD1 biofilm cultures (that are +vps1) and compare with sterile, cell-free medium. The blockage of porin permeability in the Samartzidou and Delcours study occurred when external cadaverine concentration had reached only 0.2 mM, whereas the Karatan group found 2 mM cadaverine in the external medium, suggesting that in the *V. cholerae* biofilm cells, cadaverine-blocked porins would prevent free diffusion of norspermidine out of the periplasm. Thus, cadaverine measurements would be very relevant to the authors' conclusions.

Scholarship issues:

A more thorough scholarly analysis of what is known about NspS (versus proposed here) would bolster the role for NspS in the authors' model. The current manuscript does not describe the key literature on NspS. In particular, the sensing of polyamines by MbaA through NspS-polyamine binding and direct binding of NspS to MbaA seems well-supported by literature that could be described.

The paper generally lacks context and does not discuss in adequate detail the general biological roles for polyamines (including cytoplasmically), what's already known about their roles in biofilm formation (in vibrios or other organisms like *B. subtilis*), or how they are synthesized. Without this sort of context, I think the paper will be difficult to access or appreciate for the non-expert/non-biofilm researcher.

It is surprising that the authors did not discuss and cite the Sobe et al., paper, "Spermine inhibits *Vibrio cholerae* biofilm formation through the NspS-Mba polyamine signaling system" (2017) J. Biol. Chem. 292, 17025-17036. This paper discusses the concept of self/non-self mediated by polyamines and makes a persuasive argument that spermine (a primarily eukaryotic polyamine) is a more likely signal than spermidine, especially for a human pathogen.

The authors did not measure cyclic-di-GMP directly and did not measure polyamine transport directly. It would therefore be prudent to make less categorically emphatic statements about eg., diguanylate cyclase activity line 339 "our data show that MbaA does synthesize c-di-GMP when norspermidine is present (Figure 3C, Figure 4C). In fact, c-di-GMP was not measured and the diguanylate cyclase activity of MbaA was not biochemically assayed. Similarly, for norspermidine uptake – in the abstract ("Biofilm dispersal fails in the absence of PotD1 because reuptake of endogenously produced norspermidine does not occur..") and line 109. At this stage, more tentative language would be appropriate.

Regarding self/non-self, I had difficulty integrating the discussion proposing that "self" molecules promote biofilm with a role for biofilm dispersal in cholera transmission (when, presumably, the "self" molecules would be at high abundance). I think a more sophisticated discussion of the issue here would be helpful, given that this is the lab that has defined the relevant QS signaling, has distinguished surface biofilms vs. liquid aggregates, etc.

---

## [Author Response]

Essential revisions:Experimental issues:The deletion mutants (e.g. in Figure 2) need to be complemented.

We thank the reviewers for pointing out this issue. As requested, we expressed *mbaA* and *potD1* from an ectopic locus on the *V. cholerae* chromosome. Introduction of each gene in the corresponding deletion mutant restored the WT biofilm lifecycle behavior. These results are provided in new Figure 2—figure supplement 1.

The evidence for the periplasmic sensing model in the ΔpotD1 mutant is strong, but could be further strengthened by an intervention that blocks the proposed norspermidine export. If there is a way to interrupt endogenous norspermidine synthesis and/or export to the periplasm, this would add further support for the model in Figure 5CD to explain the role of PotD1. Similarly, a PotD1 allele that is still present in the periplasm but is nonfunctional for import would provide evidence that the key issue is flux/import, and not sequestration for PotD1.

We agree that the suggested experiments would confirm our model. Unfortunately, we do not know the identity of the exporter that transports norspermidine from the cytoplasm to the periplasm, so we cannot perform that test. However, we can disrupt norspermidine biosynthesis via deletion of *nspC*, which has the added benefit of eliminating norspermidine export to the periplasm. Consistent with our model (and as has been shown previously by other groups), the Δ*nspC* strain failed to form biofilms (shown in new Figure 5F; discussed in text lines 319-327). Furthermore, we made the Δ*nspC* Δ*potD1* double mutant and it also failed to form biofilms, demonstrating that norspermidine production is epistatic to norspermidine import (shown in new Figure 5F).

To distinguish between polyamine import and PotD1-mediated sequestration of norspermidine, we deleted *potA*. PotABCD is the transport apparatus responsible for spermidine/norspermidine import. The role of the PotA ATPase is to supply the energy required for transport. Thus, in the absence of PotA, PotD1 remains capable of binding polyamines, yet transport does not occur. The Δ*potA* mutant exhibited the identical dispersal defect as the Δ*potD1* mutant validating that transport of norspermidine, not sequestration by PotD1, is the key activity required for biofilm dispersal. The results are reported in new Figure 5—figure supplement 1, along with text in lines 333-340.

The fact that the mbaA(SGDEF)* mutant still accumulates biomass at the same rate up to 8 hr as the wild-type with the same max value suggests that this pathway doesn't become relevant until later in biofilm formation. But, spermidine at high levels can inhibit early on (Figure 3A). To me, these two findings aren't reconciled in the authors' model (Figure 4F) which would lead one to believe that CDG synthesis by MbaA is critical for biofilm formation, but it's not (see Figure 4A) – to be clear, I recognize that the paper's title and abstract focus on dispersal, but the model leads one to think that the pathway being studied here is important throughout the biofilm development process. Maybe more generally, the issue is about how spermidine and norspermidine impact the normal development of a wild-type biofilm, if at all.

We thank the referees for this set of comments/suggestions. Here, we respond to them one by one for clarity:

We take the referee’s first question to be: what level of control does the MbaA signaling pathway exert over the biofilm lifecycle in WT *V. cholerae* in the absence of exogenously supplied polyamines? As the reviewer notes, in the case of the WT strain under conditions in which exogenous polyamines are not supplied, it appears, based on our results, that MbaA plays a minor role in driving progression of the biofilm lifecycle because the ∆*mbaA* mutant forms nearly WT biofilms and only exhibits a modest dispersal defect (Figure 2B). However, we have now analyzed and included results for the Δ*nspC* strain (discussed above) which is unable to synthesize and secrete norspermidine (Figure 5F). Due to the absence of endogenously-produced norspermidine, the Δ*nspC* strain does not form biofilms, as has also been demonstrated by other groups. The Δ*nspC* mutant phenotype shows that, in WT, when MbaA is present, periplasmic norspermidine is required to suppress the MbaA phosphodiesterase activity, which in turn, allows biofilms to form. Thus, in WT, the MbaA pathway has a major role as it is required for the normal biofilm lifecycle to take place.

It's not revealed until the Discussion that norspermidine isn't likely secreted at appreciable levels (something that should definitely be noted in the Introduction), so while exogenous addition can impact biofilms, the lack of secreted norspermidine makes it sound like it's actually not so critical to our understanding of WT biofilms and the behavior of WT biofilms documented in Figure 2.

As requested, we have clarified this point. We have included the following text in the Introduction (lines 92-96): “*V. cholerae* produces intracellular norspermidine, however, norspermidine has not been detected in cell-free culture fluids of laboratory grown strains (Parker et al., 2012). Thus, if norspermidine does indeed enable *V. cholerae* to take a census of ‘self’, it is apparently not via a canonical quorum-sensing type mechanism.”

In the discussion, we now write (lines 395-400), “The observation that laboratory grown WT *V. cholerae* releases little norspermidine, at least in part due to PotD1-mediated reuptake (Figure 5C), suggests that this system does not behave like a canonical quorum sensing pathway. However, it is possible that norspermidine is secreted by *V. cholerae* under some environmental conditions, or by other vibrios, and *V. cholerae* detects the released norspermidine via NspS-MbaA, and its dispersal from biofilms is prevented.”

Following on this point above, I have two questions that seem critical to assessing how polyamines impact biofilm development: (i) What's the level of spermidine produced and secreted? If the dispersal is normally driven by spermidine, its levels should accumulate extracellularly or periplasmically at the time biofilms being to disperse.

We appreciate this insight and suggestion. We performed the requested analyses. Please see lines 367-383 and the new Figure 6B. In brief, in the revised manuscript, we now provide quantitation of intracellular and extracellular spermidine and norspermidine concentrations at two timepoints – one prior to (5 h) and one after (10 h) biofilm dispersal. We find that, consistent with previous reports, *V. cholerae* makes almost no spermidine under all conditions tested. Furthermore, no substantial fluctuations in norspermidine levels (intracellular or extracellular) occurred between the timepoints that could be responsible for driving biofilm dispersal. We also considered the possibility that MbaA levels could change over the biofilm lifecycle, which could alter MbaA output activity. We therefore quantified MbaA-3xFLAG levels prior to (5 h) and after (10 h) biofilm dispersal (new Figure 6A). We found that MbaA levels remained constant. Our conclusion is that MbaA is poised to regulate the biofilm lifecycle in response to exogenous polyamines. Moreover, in the WT strain in the absence of environmentally-supplied polyamines, MbaA activity is constant and biased toward phosphodiesterase function throughout the biofilm lifecycle.

(ii) What's the biofilm phenotype of a mutant that cannot synthesize spermidine or norspermidine or both? It could be that mutants that can't synthesize polyamines behave like the WT and that neither is detectable in the periplasm/secreted fraction – such results wouldn't negate or diminish the conclusion that these polyamines can impact biofilm formation when added exogenously (which the Discussion speculates on), but it would mean that the usual pattern of biofilm mass accumulation and then drop off noted in Figure 2B isn't driven by the polyamines. At the very least, the initial accumulation seems very unlikely to be driven by norspermidine if it can't be detected.

We thank the referee for this suggestion. We have now assayed the biofilm lifecycle of the ∆*nspC* mutant that cannot synthesize norspermidine (new Figure 5F; discussed in lines 319-327). No biofilm formation occurs in this mutant demonstrating that, for normal biofilm development to occur, periplasmic norspermidine is required to suppress the MbaA phosphodiesterase activity (as discussed above in response to point 3). As noted above, we now also included a Results section demonstrating that changes in MbaA activity do not drive biofilm dispersal in the absence of exogenously supplied polyamines (new Figure 6; discussed in lines 367-383).

Following on the point above, the group of Karatan, using HPLC, has repeatedly failed to detect extracellular norspermidine in the spent growth medium of *V. cholerae* planktonic and biofilm cultures (refs. Wotanis et al., 2017 and uncited Parker et al., (2012) FEMS Microbiol Lett 329, 19-27). The current authors have used LC-MS/MS to determine that spent growth medium from planktonic potD1 gene deletion cultures contain 15 times more norspermidine than the parental wildtype strain. The authors need to provide a quantitative analysis of the absolute levels of norspermidine produced by biofilm cultures and assess whether it exceeds the minimum amount required to see effects when exogenously adding norspermidine.

We thank the reviewer for this suggestion. We have now repeated our measurements and we provide molar concentrations for all of the polyamines quantified by LC-MS/MS (see new Figure 5I, J and text lines 341-364). We write, “At high cell density, approximately 25-fold more norspermidine was present in cell-free culture fluids collected from the Δ*potD1* mutant (average 2.3 µM) than in those prepared from the WT (average 90 nM) (Figure 5I). […] There was no difference in norspermidine levels in whole cell extracts prepared from WT (average 0.6 µmol/g) and the Δ*potD1* mutant (average 0.5 µmol/g) (Figure 5J).”

Regarding our measurements from WT biofilm cultures, we now write, lines 376-379 “The concentration of extracellular norspermidine did increase between 5 h and 10 h, from <10 nM to ~75 nM (Figure 6B), however this range is far below the NspS-MbaA detection threshold (Figure 3C). Extracellular spermidine was nearly undetectable at both timepoints (Figure 6B).”

The hypothesis of the authors – that self-produced norspermidine in the Δ-potD1 mutant accumulates in the periplasm and elicits biofilm effects due to its periplasmic concentration, is elegant but requires two suppositions: firstly, norspermidine is secreted by the cell into the periplasm, and that the periplasmic concentration of norspermidine must be above a threshold that results in elicitation of biofilm and dispersal effects. Norspermidine is presumably able to freely diffuse out of the periplasm into the external medium via porins. Given the time scales involved, the concentration of norspermidine would be likely to equilibrate either side of the outer membrane. Therefore, either the concentration of norspermidine in the external cell-free medium must be above a threshold level that elicits biofilm effects, as discussed above. Or, norspermidine must be sequestered in the periplasm. How might sequestration of the norspermidine in the periplasm occur? The Karatan group showed that the cell-free spent medium from V. cholerae biofilm cultures contained 2 mM cadaverine, whereas the sterile cell-free growth medium contained only 3 μM cadaverine. This indicates that the biofilm cells were undergoing acid stress, which induces the cadaverine-producing and exporting cadAB system (lysine decarboxylase and lysine/cadaverine antiporter) but more relevantly, endogenously-produced cadaverine has been shown to block outer membrane porin channels, even if produced under pH neutral conditions (Samartzidou and Delcour (1999) "Excretion of endogenous cadaverine leads to a decrease in porin-mediated outer membrane permeability", J. Bacteriol., 181, 791-798). Production and excretion of cadaverine by CadAB during biofilm formation could prevent self-produced norspermidine diffusing out of the periplasm. In the WT cells, any periplasmic norspermidine could be reacquired by the cell through the potABCD transporter, whereas in the Δ-potD1 mutant, a cadaverine block of porins might result in accumulation of periplasmic norspermidine. Questions to the authors: (1) What was the concentration of cadaverine in the WT and Δ-potD1 cell cultures and cell-free spent medium? (2) What was the concentration of putrescine, spermidine and N-acetylnorspermidine? Presumably this data is part of the original LC-MS/MS output. (3) The authors' polyamine data was obtained with planktonic cultures containing an additional deletion of the vps1 gene that eliminates the biofilm matrix. Did the authors measure norspermidine levels in cell-free spent medium from biofilm cultures? If not, it would be very relevant to the authors' hypothesis to measure norspermidine and cadaverine in the cell-free spent medium of WT and Δ-potD1 biofilm cultures (that are +vps1) and compare with sterile, cell-free medium. The blockage of porin permeability in the Samartzidou and Delcours study occurred when external cadaverine concentration had reached only 0.2 mM, whereas the Karatan group found 2 mM cadaverine in the external medium, suggesting that in the *V. cholerae* biofilm cells, cadaverine-blocked porins would prevent free diffusion of norspermidine out of the periplasm. Thus, cadaverine measurements would be very relevant to the authors' conclusions.

We thank the referee for these insightful comments. To test the referee’s hypothesis that cadaverine blockage of porins causes norspermidine trapping in the periplasm, we performed several experiments. Our data are provided in Author response image 1. We find no evidence for cadaverine export in controlling extracellular norspermidine levels or in driving the biofilm lifecycle.

In the revised manuscript, we now provide quantitation of cadaverine, putrescine, spermidine, and norspermidine levels in all of our polyamine measurements. Those data are provided in new Figure 5 I, J; new Figure 5—figure supplement 2; new Figure 6B; and new Figure 6—figure supplement 1.

Regarding cadaverine: First, in WT planktonic, shaken cultures at high cell density, cadaverine was present at 0.5 µM in cell-free culture fluids (new Figure 5—figure supplement 2B; middle panel). In biofilm cells grown under static conditions, concentrations of both intracellular and extracellular cadaverine were low at the 5 h timepoint and dramatically increased at 10 h (new Figure 6—figure supplement 1). These results are consistent with those in the Karatan manuscript that demonstrated higher cadaverine levels for statically grown biofilm cells than for cells grown in shaken culture. Both sets of results suggest that, at late times, acid stress occurs in the static cultures, *cadAB* is induced, and cadaverine accumulates. Our temporal results, however, are inconsistent with the referee’s hypothesis that cadaverine blockage of porin channels leads to elevated norspermidine and subsequent changes in NspS-MbaA signaling. Specifically, cadaverine is in low abundance at the 5 h timepoint, when biofilm formation is favored. Cadaverine concentration increases by 10 h, after dispersal has occurred. This timing suggests that, if cadaverine had blocked porins, norspermidine would have accumulated in the periplasm by the 10 h timepoint, promoting biofilm formation, however, that does not occur. Rather, we observe biofilm dispersal.

We further examined the relationship between cadaverine and the biofilm lifecycle by constructing a Δ*cadB* mutant that lacks the lysine/cadaverine antiporter responsible for cadaverine secretion. The biofilm lifecycle of the Δ*cadB* mutant was identical to that of WT *V. cholerae* (see Author response image 1), showing that periplasmic/extracellular cadaverine does not alter norspermidine signaling in WT *V. cholerae*. Moreover, we made the Δ*cadB* Δ*potD1* double mutant and it behaved identically to the single Δ*potD1* mutant. Thus, the presence or absence of cadaverine does not alter the biofilm dispersal failure phenotype of the Δ*potD1* mutant, so the Δ*potD1* mutant phenotype cannot be due to blocking of porins by cadaverine. Together, our results show that secreted cadaverine does not impinge on the *V. cholerae* biofilm lifecycle, and therefore, cadaverine has no effect on periplasmic norspermidine levels nor on MbaA activity.

Because our results do not support any role for cadaverine in the *V. cholerae* biofilm lifecycle, we want to keep the message of our manuscript streamlined. Therefore we have not included an extended analysis of cadaverine in the current manuscript. As noted above, we do include our cadaverine measurements in the Results, and we review information on polyamines in general in the revised Introduction.

**Author response image 1. sa2fig1:** 

Scholarship issues:A more thorough scholarly analysis of what is known about NspS (versus proposed here) would bolster the role for NspS in the authors' model. The current manuscript does not describe the key literature on NspS. In particular, the sensing of polyamines by MbaA through NspS-polyamine binding and direct binding of NspS to MbaA seems well-supported by literature that could be described.

We thank the reviewers for this suggestion. We have written a more in-depth passage in the revised Introduction concerning what is known about the MbaA-NspS signaling circuit. Please see lines 78-111.

The paper generally lacks context and does not discuss in adequate detail the general biological roles for polyamines (including cytoplasmically), what's already known about their roles in biofilm formation (in vibrios or other organisms like *B. subtilis*), or how they are synthesized. Without this sort of context, I think the paper will be difficult to access or appreciate for the non-expert/non-biofilm researcher.

We regret overlooking a basic introduction to polyamines and we agree including such text will broaden the scope of the manuscript. In response, we now describe general polyamine biology in the revised Introduction (lines 69-96).

It is surprising that the authors did not discuss and cite the Sobe et al., paper, "Spermine inhibits *Vibrio cholerae* biofilm formation through the NspS-Mba polyamine signaling system" (2017) J. Biol. Chem. 292, 17025-17036. This paper discusses the concept of self/non-self mediated by polyamines and makes a persuasive argument that spermine (a primarily eukaryotic polyamine) is a more likely signal than spermidine, especially for a human pathogen.

We thank the reviewer for catching this oversight. We have included a discussion the effect of spermine on *V. cholerae* biofilm formation in our revised Introduction (lines 85-91).

The authors did not measure cyclic-di-GMP directly and did not measure polyamine transport directly. It would therefore be prudent to make less categorically emphatic statements about eg., diguanylate cyclase activity line 339 "our data show that MbaA does synthesize c-di-GMP when norspermidine is present (Figure 3C, Figure 4C). In fact, c-di-GMP was not measured and the diguanylate cyclase activity of MbaA was not biochemically assayed. Similarly, for norspermidine uptake – in the abstract ("Biofilm dispersal fails in the absence of PotD1 because reuptake of endogenously produced norspermidine does not occur..") and line 109. At this stage, more tentative language would be appropriate.

We thank the reviewer for this comment. We have now toned down our language in these and other cases.

Regarding self/non-self, I had difficulty integrating the discussion proposing that "self" molecules promote biofilm with a role for biofilm dispersal in cholera transmission (when, presumably, the "self" molecules would be at high abundance). I think a more sophisticated discussion of the issue here would be helpful, given that this is the lab that has defined the relevant QS signaling, has distinguished surface biofilms vs. liquid aggregates, etc.

As requested, we have expanded the text on the topic of polyamines as self vs non-self-signals in the Introduction, Results, and Discussion sections, with a focus on comparing the polyamine system to traditional quorum-sensing circuits. For example, see lines 391-408 for our overarching ideas concerning how the NspS-MbaA system could function to distinguish “self” vs “other.” Additionally, see lines 430-444 where we compare and contrast the opposing MbaA enzymatic activities to those of the established quorum-sensing receptor, LuxN. Beyond those new passages, at present, we hesitate to speculate on the role of MbaA signaling in disease transmission absent any data on which to hang our ideas. To our knowledge, NspS-MbaA-mediated signaling has not yet been investigated in animal models of infection.